# Contrasting carbon cycling in the benthic food webs between river-fed, high-energy canyon and upper continental slope

Chueh-Chen Tung[1], Yu-Shih Lin[2], Jian-Xiang Liao[3], Tzu-Hsuan Tu[2], James T. Liu[2], Li-Hung Lin[4], Pei-Ling Wang[1], Chih-Lin Wei[1*]

[1]Institute of Oceanography, National Taiwan University, Taipei, Taiwan
[2]Department of Oceanography, National Sun Yat-sen University, Kaohsiung, Taiwan
[3]Taiwan Power Research Institute, Taiwan Power Company, Taipei, Taiwan
[4]Department of Geoscience, National Taiwan University, Taipei, Taiwan
*Correspondence to: Chih-Lin Wei (clwei@ntu.edu.tw)

**Abstract.** The Gaoping Submarine Canyon (GPSC) off Southwest Taiwan has been extensively studied due to its unique geology, the role of transferring terrestrial material to the deep sea, and diverse biological communities. However, there is a lack of understanding of carbon cycling across the sediment–water interface in the canyon. This study aims to fill the gap by utilizing the field data collected between 2014 and 2020 and Linear Inverse Model (LIM) to reconstruct the benthic food web (i.e., carbon flows through different stocks) in the head of GPSC and the upper Gaoping slope (GS). The biotic and abiotic organic carbon (OC) stocks were significantly higher on the slope than in the canyon, except for the bacteria stock. The sediment oxygen utilization was similar between the two habitats, but the magnitude and distribution of the OC flow in the food web were distinctively different. Despite a significant input flux of ~2020 mg C m$^{-2}$ d$^{-1}$ in the canyon, 84% of the carbon flux exited the system, while 12% was buried. On the slope, 84% of the OC input (~109 mg C m$^{-2}$ d$^{-1}$) was buried, and only 7% exited the system. Bacteria processes play a major role in the carbon fluxes within the canyon. In contrast, the food web in the upper slope exhibited stronger interactions among metazoans, indicated by higher fluxes between meiofauna and macrofauna compartments. Network indices based on the LIM outputs showed that the canyon head had higher total system throughput (T..) and total system through flow (TST), indicating greater energy flowing through the system. In contrast, the slope had a significantly higher Finn cycling Index (FCI), average mutual information (AMI), and longer OC turnover time, suggesting a relatively more stable ecosystem with higher energy recycling. Due to sampling limitations, the present study only represents the benthic food web during "dry" period. By integrating the field data into a food web model, this study provides valuable insight into the fates of OC cycling in an active submarine canyon, focusing on the often overlooked benthic communities. Future studies should include "wet" period sampling to reveal the effects of typhoons and monsoon rainfalls on OC cycling.

## 1 Introduction

Submarine canyons are V-shaped valleys with steep walls carved into continental margins (Shepard, 1981). These canyons are crucial in transporting sediments and organic matter from the continental shelves to the deep ocean basins (Vetter and Dayton, 1999; Nittrouer and Wright, 1994; Epping et al., 2002; Liu et al., 2013). Turbidity currents (Shepard,1981) and mass failures (Walsh et al., 2007) are critical factors contributing to the canyon development. Therefore, the shorter the distance between the canyon head and the shore, the more efficiently terrestrial sediment and organic carbon (OC) are

transported to the deep ocean, keeping the canyon more active (Covault et al., 2007; Galy et al., 2007). However, globally, only a tiny percentage of canyons (i.e., less than 4%) are connected to the shore (Bernhardt and Schwanghart, 2021). Among the 5849 submarine canyons in the world, only 2.79% are river-fed (Harris and Whiteway, 2011). They are more prevalent along the active margins with narrow shelves and onshore river catchments, likely promoting the canyon incisions through water discharges and coarse-grained bedload erosions (Bernhardt and Schwanghart, 2021). Although globally rare, the shore and river-connected canyons dominate Taiwan's continental margin (seven out of the 13 submarine canyons; Chiang and Yu, 2022). Among them, the Gaoping submarine Canyon (GPSC) is the largest and the only river-fed canyon in the southwestern (SW) Taiwan margin (i.e., among Penghu, Shoushan, Kaohsiung, Gaoping, and Fangliao Canyons, and Hongtsi Sea Valley, Chiang et al., 2020), providing a unique opportunity to investigate the shore- and river-connected canyon ecosystem.

The GPSC is deeply incised into the Gaoping Shelf. The canyon head is ~1 km away from the Gaoping River (GPR), a small mountainous river (SMR). The GPR drainage basin has steep elevation changes (Liu et al., 2016) and is frequently disturbed by earthquakes and typhoons (Liu et al., 2009). As a result, the GPR discharges a substantial amount of sediment to the GPSC each year (45.6 to 110 Mt; Taiwan Water Resources Agency), approximately 30 to 80% of the Mississippi River's yearly export (Meade and Moody, 2010). The frequent earthquakes and typhoons also lead to turbidity currents and gravity flows, causing continuous erosion of the canyon floor and sediment mobilizations (Gavey et al., 2017; Ikehara et al., 2020; Chiang et al., 2020).

The riverine carbon fluxes in the GPR–GPSC system vary with wet and dry seasons. The wet season occurs from summer to early autumn due to the impact of monsoons and typhoons, contributing about 78% of the yearly charge of the river (Liu et al., 2002) and up to ~46 times more particulate organic carbon (POC) transport than in the dry season (Liu et al., 2016). Hsu et al. (2014) showed that the sedimentary OC accumulation rates in the Gaoping shelf and slope area account for less than 13% of the riverine POC load, suggesting that most POC likely exits the GPSC and is buried in the deep South China Sea (SCS) (Hsu et al., 2014; Kao et al., 2014; Liu et al., 2013; 2016). However, these earlier studies did not consider the impact of benthic organisms, which play a quantitatively important role in oxidizing sedimentary OC through feeding, respiration, burrowing, and predation activities. Therefore, OC cycling and budgets of the source-to-sink system cannot be accurately reconstructed without including benthic organisms (Snelgrove et al., 2018).

Submarine canyons are known to transport a greater quantity and quality of sedimentary OC than sloping environments at a similar depth (Garcia et al., 2007; Pusceddu et al., 2010; Vetter and Dayton, 1999), thereby enhancing carbon oxidation rates (Epping et al., 2002; Rabouille et al., 2009), benthic standing stocks (Ingels et al., 2009), and deposit-feeding (Amaro et al., 2009; De Leo et al., 2010; Vetter and Dayton, 1999). These processes indicate intensified carbon cycling in the canyon benthic community. However, only a few studies have adopted the food web modeling approach to identify the key benthic size classes responsible for OC oxidation as well as the magnitude, turnover, and structure of carbon stocks and flow in the submarine canyons (Rowe et al., 2008; van Oevelen et al., 2011). It is noted that these observations were made based on studies of submarine canyons on passive margins.

For submarine canyons on active margins, our previous work showed that strong physical disturbance in the shore-connected GPSC had significant negative impacts on benthic community structure and function, resulting in lower abundance, diversity, biomass, growth, respiration, and distinctly different composition compared to the nearby slope ecosystem (Liao et al., 2017, 2020; Tung et al., 2023). However, the carbon flows within the canyon's benthic food web in

active margins, including that of the GPSC, remain underexplored. To address this gap, we applied Linear Inverse Models (LIM) to reconstruct and compare the carbon flows within the food webs of the head of GPSC and the nearby Gaoping Slope (GS). We analyzed the food web characteristics, quantified the carbon flow, and discussed how the unique biological communities in the high-energy GPSC affect carbon cycling and budget. We hope to close the knowledge gap on the carbon flow in a dynamic submarine canyon setting on an active margin.

## 2 Materials and Methods

### 2.1 Characteristics of Gaoping Submarine Canyon and Gaoping Slope

The upper GS (about 200–300 m water depth) experiences relatively high sediment accumulation (Hsu et al., 2014), while the GPSC is susceptible to surface sediment erosion triggered by turbidity currents (Mulder et al., 2001), gravity flows (Liu et al., 2013), and powerful internal tides originating from the Luzon and southeastern Taiwan Strait (Chiou et al., 2011; Jan et al., 2008; Lee at al., 2009a, b). The convergence of internal waves into beam patterns along the canyon thalweg generates bottom-intensified currents (Wang et al., 2008), whereas the presence of isopycnal surfaces from Gaoping River plumes within the canyon head facilitates the generation and propagation of internal tides (Lee et al., 2009; Liu et al., 2002; Wang et al., 2008). Consequently, the upper GPSC (about 200–400 m water depth) seafloor features a benthic nepheloid layer (BNL) with a thickness exceeding 100 m and a suspended sediment concentration surpassing 30 mg $l^{-1}$, sustained by current-induced resuspension related to internal tides (Liu et al., 2002; 2010; Liu and Lin, 2004). The impacts of the physical disturbance on the sedimentary OC and benthos in the canyon include (*i*) the removal of fine particles containing higher total organic carbon (TOC) content, resulting in diminished food supplies and subsequently lower meiofaunal and macrofaunal abundance, biomass, and diversity in the canyon compared to the adjacent slope (Liao et al., 2017; 2020; Tung et al., 2023), and (*ii*) a significant negative impact on the nematode functional and trophic diversity, as well as community maturity (Liao et al., 2020), affecting various aspects of the benthic ecosystem functions, such as size structure, community growth, and respiration (Tung et a., 2023).

### 2.2 Sampling

Between 2014 and 2020, the sampling stations at upper GPSC and GS (abbreviated as GC1 and GS1) were repeatedly surveyed by the National Taiwan University's RV *Ocean Researcher 1* and *New Ocean Researcher 1* (Fig. 1). During each visit, we used a CTD/rosette for the hydrographic survey and either a UNSEL box corer (Hessler and Jumars, 1974) or an OSIL megacorer for sediment sampling. Seawater temperature and salinity were measured with a CTD recorder (Sea-Bird SBE 911). For the box core operation, five transparent polycarbonate tubes (i.d.= mm) were inserted into the sediments to take subsamples. For the megacorer operations, a maximum of 12 polycarbonate tubes (i.d.= 105 mm) were recovered to obtain replicate samples. The cruise details, sampling sites, and sampling gears are listed in Table A1.

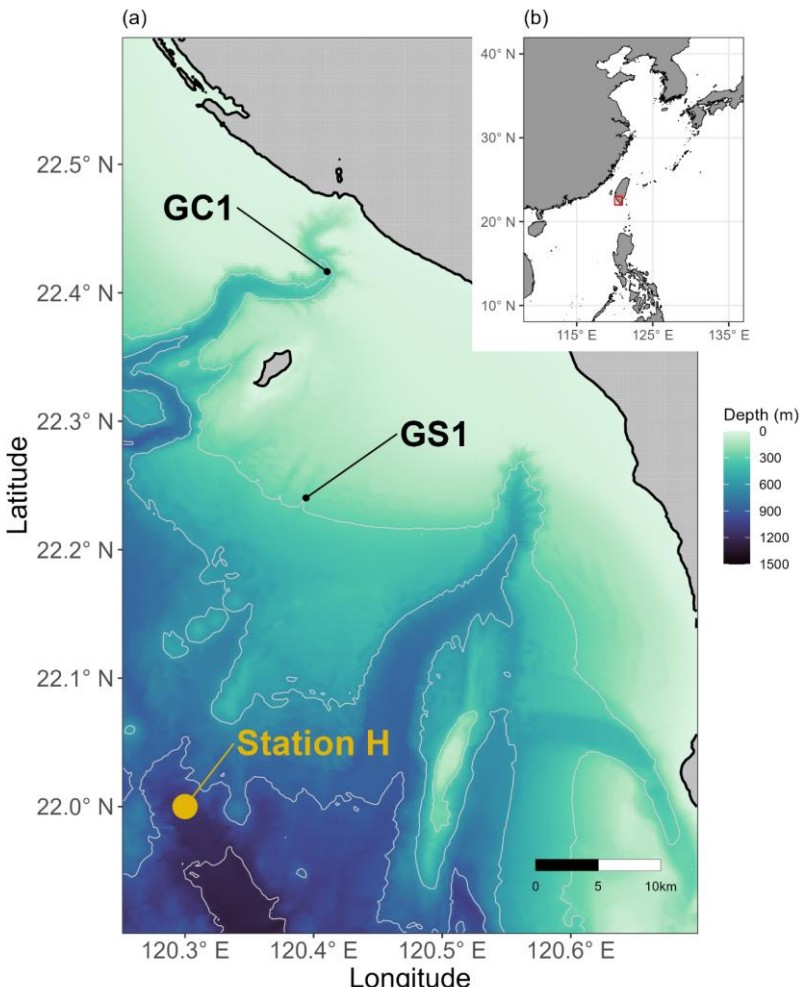

**Figure 1. Map of sampling stations visited from 2014 to 2020. (A) Sampling sites at the head of Gaoping Submarine Canyon (GPSC) and the upper Gaoping Slope (GS), are abbreviated as GC1 and GS1, respectively. POC fluxes collected by drifting sediment trap at station H (Shih et al. 2020) adjacent to GS1 are also annotated. (B) The sampling area off the continental margin of SW Taiwan.**

## 2.3 Food web structure

This study utilized Linear Inverse Models (LIM) to establish connections between the abiotic and biotic components within the food webs of GC1 and GS1 (illustrated in Fig. B1). In the model, the influx of POC consists of a mixture of OC derived from the water column, with a portion exiting the network through burial and export processes such as removal from the sediment surface or resuspension by bottom currents (represented by orange flows). The faunal compartments are categorized based on size classes, encompassing bacteria (BAC), meiofauna (MEI), and macrofauna (MAC). A simplified

microbial loop is included, with a portion of detritus degrading into a dissolved organic matter (DOC) pool, which bacteria then take up. Bacteria die and contribute to the DOC pool through viral lysis. The predators from each size class consume organisms of all smaller size classes (black flows). For example, meiofauna only consumes detrital OC and bacteria. Macrofauna preys on meiofauna, bacteria, and detrital OC. Carbon loss includes fauna mortality and defecation (i.e., flow back to detritus stock), consumption by benthopelagic/pelagic predators, and transfer to DIC through respiration (blue

flows in Fig. B1). All food web components are linked by mass-balance equations, describing the biomass changes due to

the difference between feeding and carbon loss (i.e., feces, respiration, mortality, and consumption). The carbon flows between components are constrained by inequality equations, which use the site-specific or physiological upper and lower bounds from direct measurements, theoretical calculations, or literature values. Since almost all sampling was conducted in autumn and spring (i.e., only one cruise in summer, Table A1), the LIM food webs only used the "dry" period data (i.e., autumn and spring).

### 2.4 Data sources

### 2.4.1 Sampling procedures of non-living component of OC

Surface sediment (0–1 cm) was collected by a 50 ml centrifuge tube and preserved in a -20°C freezer. The sediment samples were freeze-dried for 3 to 5 days to measure wet weight (before freeze-drying), dry weight (after freeze-drying), water content, and porosity. An aliquot of freeze-dried sediment (~0.4 g) was acidified with HCl to remove carbonate, combusted at 1000°C with pure oxygen, and analyzed with a Flash EA 1112 elemental analyzer for TOC. The core area ($m^2$) was first converted to volume ($m^3$) by multiplying the depth at which the macrofauna may dwell (i.e., 10 cm). The volume was then converted into mass using the sediment dry densities converted from the bulk densities reported by Su et al. (2018), which was 2.1 (g $cm^{-3}$) in GC1 and 1.8 (g $cm^{-3}$) in GS1, corrected by the water contents (Table A2). Finally, the OC stock was determined by multiplying the sediment mass with TOC content (%) and dividing by the sampling area, resulting in the unit of mg C $m^{-2}$. Because only a small aliquot of sediment was used in TOC measurement, the sample likely excluded larger fauna and was assumed to contain only detrital OC.

### 2.4.2 Sampling procedures of living component of OC

Prokaryotes in the surface sediment (0–1 cm) were taken from a cut-off 10 ml sterile syringe (i.d. 15 mm) within a core tube. A 2 ml subsample was added to a 15 ml polyethylene centrifuge tube containing 2 ml of pre-filtered PBS solution, and the mixture was then fixed in 2% paraformaldehyde. Based on the procedures described in Kallmeyer et al. (2008), prokaryotic cells in the fixed samples were detached from the sediments. The detached cells were subsequently stained with SYBR Green and DAPI, and mounted on a slide for enumeration (Deming and Carpenter, 2008). A total of 10 slides were prepared for each station to calculate the mean prokaryotic abundance. The stock of prokaryotic OC was calculated by assuming a conversion factor of 10 fg C per cell (Deming and Carpenter 2008) and converted to the unit of mg C $m^{-2}$. Meiofauna was collected from the top 5 cm of sediment using a cut-off 10 ml sterile syringe (i.d. 15 mm) (Montagna et al., 2017). Three samples in each station (from different core tubes) were fixed in 5% formalin, stained with Rose Bengal, and wet sieved through a 40 µm sieve before transferring to 70% ethanol. After centrifuging at 8,000 rpm for 10 min in Ludox HS40 solution (repeated three times) (Danovaro, 2009; Montagna et al., 2017), the meiofauna specimens were enumerated into major taxonomic groups under a high-power stereomicroscope (Olympus® SZX16; 0.7–11.5 X zoom). Both the macrofauna and meiofauna were identified to the lowest possible taxonomic level (phylum, class, or order).

At least three core tubes were retained in each station for macrofauna analysis. The top 10 cm of the sediments (including supernatant water) were wet-filtered through a 300 µm sieve (Montagna et al., 2017). After being fixed in 5% formalin and stained with Rose Bengal, the macrobenthos samples were sorted and enumerated into major taxonomic groups and then permanently preserved in 70% ethanol.

The body volume of meiofauna and macrofauna specimens was calculated using the formula:

$$V = L \times W^2 \times C, \tag{1}$$

where $V$ is the volume, $L$ is the length, $W$ is the width, and $C$ is the taxon-specific conversion factor (Warwick and Gee, 1984). For the taxa without conversion factors, the biovolume was calculated from length and width using the nearest geometric shapes (e.g., cone shape: scaphopods; cylinder shape: aplacophorans, sipunculans, and nemerteans; ellipsoid shape: ophiuroids and asteroids) (Tung et al., 2023). The biovolume was first converted into wet weight by assuming a specific gravity of 1.13 (Warwick and Gee, 1984). Subsequently, it was converted into OC using the conversion factor of 12% for meiofauna (Baguley et al., 2004) and 4.3% for macrofauna (Rowe, 1983). Finally, all the OC stock of fauna was divided by the sampling area and then converted to the unit of mg C m$^{-2}$. The size data exceeding three standard deviations from the mean (i.e., z-score) were excluded as outliers before estimating the total biomass.

### 2.4.3 Sediment Community Oxygen Consumption (SCOC)

Three core tubes recovered from the megacorer were used to measure SCOC for each station. The incubation was conducted in a dark, temperature-controlled water bath. If the supernatant water was insufficient to fill the core tube, the sediment core was carefully filled with bottom water collected from the CTD rosette. After carefully removing air bubbles, the core tube was sealed hermetically with a custom-built HDPE lid attached to a magnetically driven impeller (60–80 rpm) to circulate the water during the incubation. Then, the sediments were acclimated for approximately 6 hours until the flocculent materials settled and the overlying water was clear. The dissolved oxygen concentration was measured every 8 hours with a miniature oxygen optode (i.d. 2 mm) through a sampling port on the core lid (PreSens® Microx 4). According to Glud (2008), the dissolved oxygen concentration was measured until it decreased by 15% of the initial concentration to prevent hypoxic stress. The fluxes of oxygen into or out of the sediments were calculated as

$$\text{Flux} = [\text{Change in concentration}] \times \frac{V}{A} \times T \tag{2}$$

Here $V$ is the volume of the overlying water, $A$ is the core area, and $T$ is the incubation time.

After shipboard incubation, one to three oxygen microelectrodes (100 µm tip size) were inserted simultaneously into sediments at 100 µm increments using Unisense® Field Microprofiling System. The diffusive oxygen fluxes through the sediment–water interface were calculated according to Fick's first law of diffusion (Berg et al., 1998; Glud, 2008). The oxygen concentration profile, sediment porosity, initial concentrations in the overlying water, and oxygen diffusion coefficient corrected by temperature were input to calculate diffusive oxygen fluxes using Unisense® Profile software. Diffusive carbon remineralization was estimated from the oxygen flux by assuming a respiratory quotient (RQ) of 0.85 (Jørgensen et al., 2022).

Sediment oxygen profile concentration measures the diffusive oxygen utilization (DOU) mainly contributed by the aerobic respiration of microorganisms through the slow diffusion of oxygen molecules and the chemical oxidation of upward diffused reduced species (e.g., $NH_4^+$, $Mn^{2+}$, $Fe^{2+}$). In contrast, the sediment incubation experiment measures total oxygen utilization (TOU). TOU not only accounts for DOU but also benthos' respiration and the benthos-mediated oxygen utilization (BMU) through their bioirrigation and bioturbation activities (Glud, 2008; Lichtschlag et al., 2015; Wenzhöfer and Glud, 2004). Therefore, we defined the difference between TOU and DOU as BMU, characterizing benthos' contribution to sediment oxygen dynamics.

### 2.4.4 Geochemical flux constraints

Due to the difficulty in accurately partitioning TOU among the OC stocks (Glud, 2008), we used DOU as the geochemical constraint instead of TOU, as the former can be associated explicitly with BAC→DIC (Fig. B1). However, instead of directly using the measured DOU as input data, we established the upper bound of BAC→DIC at 30% of the measured TOU, according to Mahaut et al. (1995) (Table 1). This chosen method enhances model flexibility. The measured TOU, DOU, and calculated BMU are used to validate or compare the modeled results.

Extensive sediment accumulation rate (SAR) data are available in our study area from Tsai and Chung (1989), Huh et al. (2009), and Su et al. (2018). Therefore, the OC burial rate was estimated by multiplying the SAR with TOC content (Jahnke, 1996). Carbon burial efficiency (Epping et al., 2002) was calculated through

$$\text{Carbon burial efficiency} = \frac{\text{Burial rate}}{\text{POC flux}}. \qquad (3)$$

Equation (3) requires POC flux, which is constrained by the minimum and maximum values of the sediment trap data before spring tides in GC1 (Liu et al., 2006; 2009). The POC flux measurements are not available in GS1, but POC fluxes collected by drifting sediment trap at station H (Fig. 1) adjacent to GS1 are reported by Shih et al. (2020). Therefore, the range of POC fluxes (Shih et al., 2020) below the euphotic zone divided by trapping efficiency (Hung and Gong, 2007; Li, 2009) are used as the POC flux constraints in GS1.

Considering that the accumulated sediment budget can only account for 13–18% of GPR's annual sediment load, Huh et al. (2009) suggested that gravity flows through the GPSC may export the remaining sediment. Therefore, the upper bound of the sediment export rate is set to 87% at both sites (Table 1). The geochemical constraints are listed in Table 1, and the derivations from the literature are summarized in Table A2.

**Table 1. Equality and inequality constraints implemented in the LIM models of GC1 and GS1. Values given as single numbers implied that the data were equalities, whereas the values designated in [minimum value, maximum value] represented the inequalities.**

| Inequality description | Calculation | GC1 | GS1 | Unit | Reference |
|---|---|---|---|---|---|
| Temperature limitation (*Tlim*) | $\mathbf{Q10 \times exp(((T-20))/10)}$, with Q10 = 2 | 1.05 | 1.09 | - | |
| Bacteria growth efficiency | | [0.02, 0.61] | [0.02, 0.61] | - | del Giorgio and Cole (1998) |
| Virus-induced prokaryotic mortality | $\mathbf{14.46 \times ln(x) - 24.98}$, with x= water depth | [0,0.62] | [0,0.57] | | Danovaro et al. (2008) |
| DOU (BAC → DIC in Fig. B1) | **Maximum DOU = 30% TOU** | [0.0, 20.87] | [0.0, 15.526] | mg C m$^{-2}$ d$^{-1}$ | Mahaut et al. (1995) |
| Maintenance respiration of meiofauna | | | | mg C m$^{-2}$ d$^{-1}$ | van Oevelen et al. (2011) |
| Maintenance respiration of macrofauna | $\mathbf{Tlim \times 0.01 \times Stock}$ | | | mg C m$^{-2}$ d$^{-1}$ | van Oevelen et al. (2011) |
| Assimilation efficiency of meiofauna | | [0.456, 0.699] | [0.456, 0.699] | - | Conover (1966) |
| Assimilation efficiency of macrofauna | | [0.6 0.7] | [0.6 0.7] | - | Loo and Rosenberg (1996) |
| P/B ratio of meiofauna | | [0.0090, 0.0493] | [0.0090, 0.0493] | - | Fenchel (1982); Fleeger and Palmer (1982) |
| P/B ratio of macrofauna | | [0.0008, 0.0048] | [0.0008, 0.0048] | - | Stratmann et al. (2018) |
| Mortality of meiofauna | | $\mathbf{Tlim \times [0, 0.20] \times Stock}$ | | mg C m$^{-2}$ d$^{-1}$ | Hendriks (1999); van Oevelen et al. (2012) |
| Mortality of macrofauna | | $\mathbf{Tlim \times [0, 0.05] \times Stock}$ | | mg C m$^{-2}$ d$^{-1}$ | Hendriks (1999), Tenore (1982); van Oevelen et al. (2012) |
| Net growth efficiency of meiofauna | | [0.3, 0.5] | [0.3, 0.5] | - | Herman and Heip (1985); Banse and Mosher (1980); Herman et al. (1983; 1984) |
| Net growth efficiency of macrofauna | | [0.6, 0.72] | [0.6, 0.72] | - | Navarro et al., (1994); Nielsen et al., (1995) |

| Inequality description | Calculation | GC1 | GS1 | Unit | Reference |
|---|---|---|---|---|---|
| POC flux | | [800, 2400] | [52.5, 153.3] | mg C m$^{-2}$ d$^{-1}$ | Liu et al. (2006); Shih et al (2020); Hung and Gong, (2007); Li (2009) |
| Burial efficiency | **Burial rate/POC flux** | [0.067, 0.203] | [0.66, 1] | - | Tsai and Chung (1989); Huh et al. (2009) |
| Export rate | | [0.82, 0.87] | [0, 0.87] | - | Huh et al. (2009) |

### 2.4.5 Physiological constraints

We applied the four wildly used physiological constraints in LIM studies (van Oevelen et al., 2006; Stratmann et al., 2018) in the food web model, including respiration (R), assimilation efficiency (AE), production (P), and net growth efficiency (NGE).

For meiofauna and macrofauna, R is defined as the sum of maintenance respiration (biomass-specific respiration, MR) and growth respiration (associated with growth processes, e.g., synthesis of new structures in growth, GR). MR is set to be proportional to 1% of the biomass per day at 20˚C following the definition in van Oevelen et al. (2006), corrected by a temperature correction factor (*Tlim*) (Soetart and van Oevelen, 2009) as

$$Tlim = Q10 \times \exp\left(\frac{(T-20)}{10}\right). \tag{4}$$

Here Q10 is a coefficient for temperature-dependent reaction and was set to 2 for most poikilotherms. *T* is the bottom water temperature for each site. Therefore, the maintenance respiration is calculated as

$$MR = 0.01 \times Tlim \times \text{Stock.} \tag{5}$$

Within the food web model, GR is defined as

$$GR = R - MR. \tag{6}$$

AE is described as

$$AE = \frac{(I-F)}{I}. \tag{7}$$

Here I is the ingested food, and F is the feces.

P and P/B ratio are expressed as

$$P = I - F - GR, \tag{8}$$

$$\text{and } \frac{P}{B} = \frac{(I-F-GR)}{\text{Stock}}, \tag{9}$$

respectively. Bacterial growth efficiency (BGE) is defined as the amount of new bacterial biomass produced per unit of assimilated OC (del Giorgio and Cole, 1998). Net growth efficiency (NGE) (Clausen and Riisgård, 1996) is defined as

$$NGE = \frac{(I-F-GR)}{(I-F)} = \frac{P}{(P+GR)}. \tag{10}$$

The AE, P/B ratio, BGE, and NGE constraints are derived from the literature and summarized in Table 1.

### 2.5 Linear inverse models

With a priori food web structure (Fig. B1), the empirical data of standing stocks, geochemical flux constraints, and physiological constraints were combined as a set of linear functions of equality and inequality equations (van Oevelen et al., 2010). Several reviews have explained the technical and methodological aspects of LIM (e.g., van Oevelen et al., 2006; 2010; Soetaert and van Oevelen, 2009), thus not rehashed here. We adopted a likelihood approach based on Markov Chain Monte Carlo (MCMC) algorithm (Kones et al., 2006) to calculate the possible solution sets for LIM. The standard deviation is set to a ± 2% error margin to iterate until the convergence of solution sets. The model was run 10,000 times to estimate the mean of unknown flows using R-package LIM (Soetaert and Herman, 2009; van Oevelen et al., 2010).

**2.6 Turnover rate of OM**

The turnover rate (or residence time) of OM is a function of stock size and energy transfer in/out of stock (Rowe et al., 1990; 2008). Therefore, the OC turnover rate was quantified by dividing the carbon stock with measured oxygen consumption or modeled carbon flows. For example, the OC turnover rate was calculated as the sum of detrital and biotic carbon divided by measured TOU or by the sum of flows BAC→DIC, MEI→DIC, and MAC→DIC, corresponding to the definition of TOU.

**2.7 Network indices of ecosystems**

Food web functions were examined using network indices widely applied in deep-sea food web studies, including total system throughput (T..), total system throughflow (TST), total system cycled throughflow (TST$_C$), Finn's cycling index (FCI), and average mutual information (AMI) (Table A3 and A4, Latham and Scully, 2002; Ulanowicz, 2004; Kones et al., 2009). All the indices were calculated using R-package NetIndices (Kones et al., 2009). The fraction of indices (from 10,000 solutions) with greater values in the GC1 than GS1 was calculated to compare the difference between sites. For instance, when the fraction is 0.9, it implies that 90% of the solutions were greater in the GC1 than GS1, and vice versa for a fraction of 0.1. The difference was considered significant when the fraction was greater than 90% or smaller than 10%, and highly significant when the fraction was greater than 95% or smaller than 5% (van Oevelen et al., 2011).

**2.8 Statistical analysis**

Seasonality of OC stock and oxygen utilization was first examined to determine whether the sampling seasons may affect the analysis. The assumption of homogeneous variance was examined by permutational analysis of dispersion (PERMDISP). Mixed effect permutational analysis of variance (PERMANOVA) was used to investigate the impact of habitat (canyon v.s. slope) and season (spring, summer, and autumn) on the biotic, abiotic carbon stocks and oxygen utilization. The number of permutations for each test was set to 9999, and all statistical tests used α-value = 0.05. The multivariate statistical analyses used R-package "vegan" (Oksanen et al., 2022).

**3 Results**

**3.1 Environment data**

The average bottom water temperature was 13.5 °C in GC1 and 13.9 °C in GS1. As a result, the calculated *Tlim* are 1.05 and 1.09 for GC1 and GS1, respectively. *Tlim* values are important correction factors controlling biomass-specific maintenance respiration in the LIM model. While the bottom water density, salinity, and temperature were comparable between GC1 and GS1, the light transmission profiles showed a notable distinction (Fig. B2). For instance, little to no light transmission was observed near the GC1 seafloor throughout the sampling cruises, indicating the persistent presence of BNL. This result coincided with the observation of Liu et al. (2010), who reported a 100 m thick BNL with high suspended sediment concentration in GPSC based on mooring observations.

### 3.2 Stock of the non-living component of OC

For the abiotic sediment OC stock, the average TOC content in the upper 10 cm of the sediment was $0.4 \pm 0.11\%$ and $0.5 \pm 0.04\%$ in GC1 and GS1, respectively. This value fits in the range of TOC content (c.a. 0.3–0.8%) reported previously in the southwestern margin of Taiwan (Kao et al., 2006; Hsu et al., 2014). PERMDISP showed that the assumption of homogenous variation is not violated between habitat groups and among season groups (Table A5, Habitat, p= 0.4469; Season, p= 0.9375). The sediment OC stock in the GS1 was significantly higher than in the GC1 (Table A6, PERMANOVA, p= 0.0399), but the seasonal variability was not evident (p= 0.8077). Since the seasonal signal was not detected, we used the average dry-period OC of $442{,}915 \pm 88{,}074$ mg C $m^{-2}$ for GC1 and $540{,}985 \pm 93{,}128$ mg C $m^{-2}$ for GS1 in the LIM model (Table 2, Fig. 2a).

**Table 2. Mean standing stocks (in mg C $m^{-2}$) with standard deviation from GC1 and GS1 in the dry period.**

|  | Source | GC1 | GS1 |
|---|---|---|---|
| Carbon stock (mg C $m^{-2}$) | Sediment | $442{,}915 \pm 88{,}074$ | $540{,}985 \pm 93{,}128$ |
|  | Bacteria | $65.3 \pm 12.74$ | $42.8 \pm 6.75$ |
|  | Meiofauna | $0.69 \pm 0.90$ | $46.41 \pm 19.60$ |
|  | Macrofauna | $2.1 \pm 1.94$ | $50.03 \pm 43.90$ |

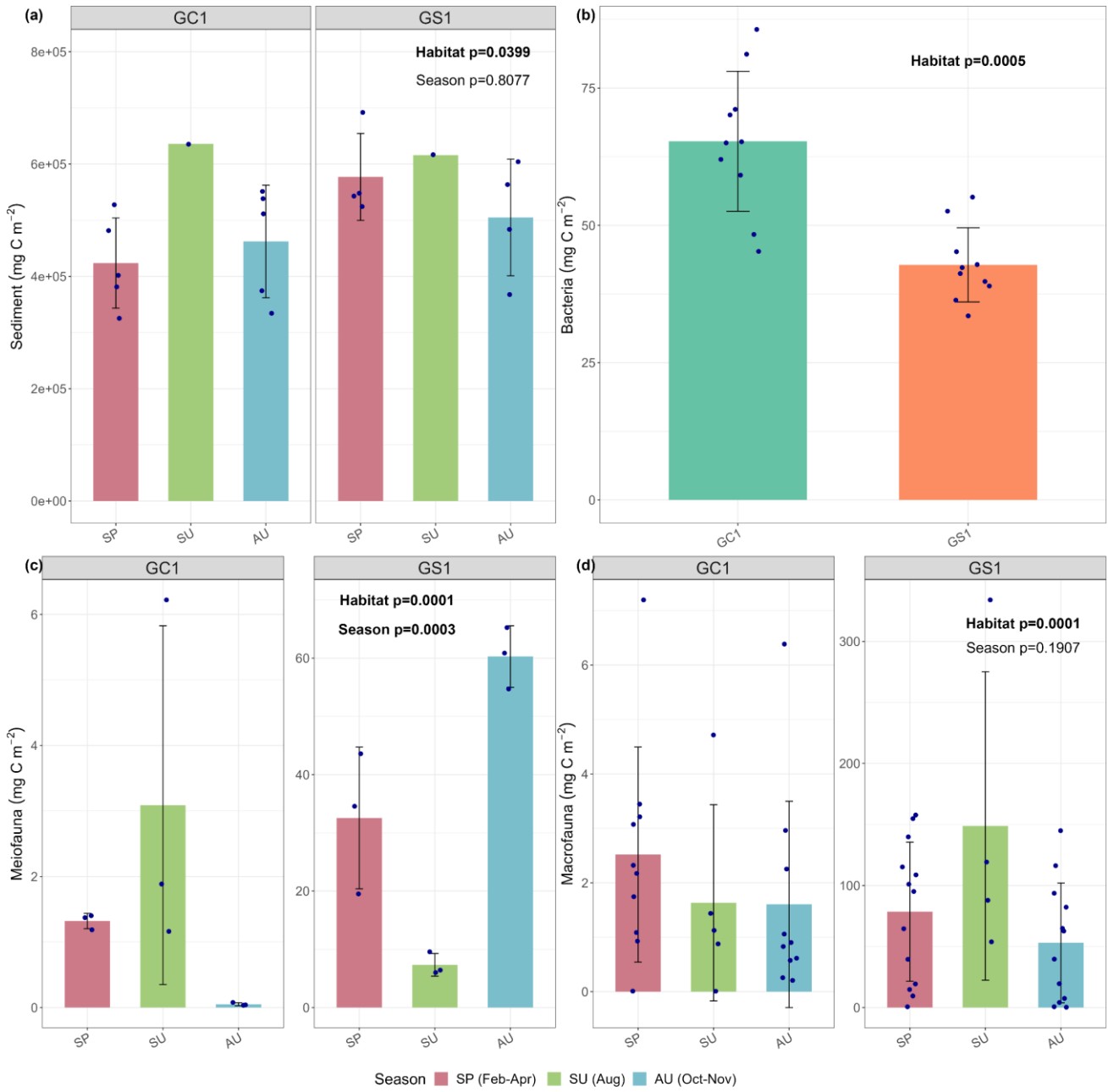

**Figure 2. Carbon stocks of (a) sediment, (b) bacteria, (c) meiofauna, and (d) macrofauna of GC1 and GS1. The bar chart and error bars show the mean and standard deviation of carbon stocks. The dots represent the OC stock (in mg C m⁻²) for each replicate. Note that summer sampling of sediment was only conducted once in Aug 2015 (cruise OR1_1114). Bacteria was only collected in Mar–Apr 2018 (cruise OR1-1190). Meiofauna was only collected once each season for both sites, so the three replicates were conducted during a single cruise.**

## 3.3 Stocks of the living component of OC

Prokaryotes were only sampled in spring 2018 (cruise OR1_1190) with ten replicates in each site. Prokaryote biomass in GC1 was significantly higher than in GS1 (Fig. 2b, Table A6, PERMANOVA, p= 0.0005). For the model input, the average bacteria biomass is set to 65.3 ± 12.7 mg C m⁻² in GC1 and 42.8 ± 6.8 mg C m⁻² in GS1, respectively (Table 2).

PERMDISP indicates that the meiofaunal variation between habitat and season groups was not homogeneous (Table A5, Habitat, p= 0.0001; Season, p= 0.0001). The meiofaunal biomass in GS1 was significantly higher than in GC1 (Table A6, PEMANOVA, p= 0.0002) with significant seasonal variability (p= 0.006), but also a significant interaction between the season and habitat (p= 0.0002). Due to the lack of consistent seasonal pattern between habitats and only one sampling cruise was conducted for each season (Fig. 2c), the average meiofaunal biomass in the dry period was used in the model, with $0.69 \pm 0.90$ mg C m$^{-2}$ in GC1 and $46.41 \pm 19.60$ mg C m$^{-2}$ in GS1.

Macrofauna sampling was conducted for eight cruises across the two habitats. Macrofaunal biomass was approximately an order of magnitude lower, and abundance was about five times lower in the GC1 than in GS1 (Fig. 2d). PERMDISP showed a significant difference in variance between habitat groups but not among season groups (Table A5, PERMDISP, Habitat, p= 0.0001; Season, p= 0.3136). In addition, there was a significant biomass difference between habitats but not among seasons (Table A6, PERMANOVA, Habitat, p= 0.0001; Season, p= 0.1907). Therefore, the average macrofaunal biomass in the dry period was considered in the LIM model, with $2.1 \pm 1.94$ mg C m$^{-2}$ for GC1 and $50.03 \pm 43.90$ mg C m$^{-2}$ for GS1 (Table 2).

Table 2 summarizes the mean standing stocks used for the LIM model inputs. All OC stocks were higher in GS1 than in GC1, except for the bacteria. The stock of non-living components was approximately four orders of magnitude higher than that of living components. The meiofauna and macrofauna biomass were remarkably depressed in the canyon due to physical disturbances (Liao et al., 2017; 2020; Tung et al., 2023). Macrofauna had greater biomass than meiofauna on both the slope and canyon.

### 3.4 Oxygen utilization

The variation and mean of all oxygen utilization measurements (TOU, DOU, and BMU) were not statistically different among seasonal groups (Fig.3; Table A5, PERMDISP, p>0.05) and between habitat groups (Fig.3; Table A6, PERMANOVA, p>0.05). The means and standard deviations of TOUs were $69.6 \pm 39.9$ mg C m$^{-2}$ d$^{-1}$ for GC1 and $51.8 \pm 15.2$ mg C m$^{-2}$ d$^{-1}$ for GS1 (Fig.3; Table 2). The measured DOUs representing aerobic respiration of bacteria showed little difference between the two sites, with values of $14.4 \pm 15.8$ mg C m$^{-2}$ d$^{-1}$ in the canyon head and $9.0 \pm 9.5$ mg C m$^{-2}$ d$^{-1}$ on the slope (Fig.3; Table 2), despite the higher bacterial biomass in GC1 than GS1. Note that the standard deviations were greater than or close to the mean value in both sites, suggesting wide variations existed between samplings. BMU was presumed lower in GC1 due to lower metazoan biomass and thus lower benthos-mediated oxygen utilization. Nevertheless, BMU was $49.0 \pm 28.2$ mg C m$^{-2}$ d$^{-1}$ in GC1, and $41.2 \pm 12.3$ mg C m$^{-2}$ d$^{-1}$ in GS1 (Fig.3; Table 2), showing no statistical difference between the two habitats.

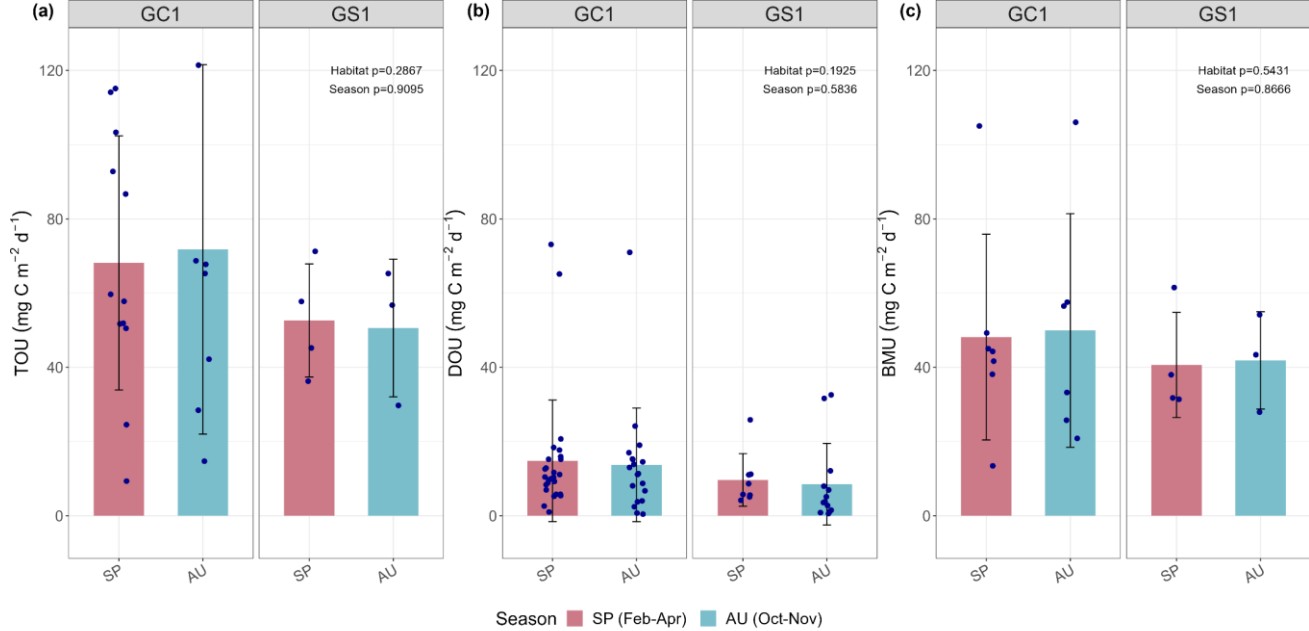


**Figure 3. Measured (a) total oxygen utilization (TOU), (b) diffusive oxygen utilization (DOU), and (c) benthos-mediated oxygen utilization (BMU) of GC1 and GS1. The bar chart and error bars show the mean and standard deviation for spring and autumn cruises. The dots represent the measurements for each replicate. The dotted lines represent the mean values from all sampling.**

**No summer data is available.**

### 3.5 LIM results

MCMC-solved flows in the GC1 model can be separated into three parts according to the order of magnitude of the mean values (Fig. 4a and Table A7). First, the mean value for POC flux was 2017.5 mg C m$^{-2}$ d$^{-1}$. The export (1696.6 mg C m$^{-2}$ d$^{-1}$) occupied 84% of the POC input. Although 16% of the POC flux (320.8 mg C m$^{-2}$ d$^{-1}$) retains in the sediment, only 4%

(72.4 mg C m$^{-2}$ d$^{-1}$) enters the benthic food web. The second-largest flow values were those related to the microbial loop, including DOC→BAC (26.3 mg C m$^{-2}$ d$^{-1}$), BAC→DOC (12.4 mg C m$^{-2}$ d$^{-1}$), and BAC→DIC (13.8 mg C m$^{-2}$ d$^{-1}$). Although the remaining flows were less than 1 mg C m$^{-2}$ d$^{-1}$, the meiofauna-related flows were at least one order of magnitude higher than the macrofauna-related flows.

The estimated POC flux input was 103.0 mg C m$^{-2}$ d$^{-1}$ in the GS1 by MCMC method, with about 7% (8.6 mg C m$^{-2}$ d$^{-1}$)

exported out of the system (Fig. 4b). While 93% of the POC flux (121.5 mg C m$^{-2}$ d$^{-1}$) is retained in the sediment, 10% (12.6 mg C m$^{-2}$ d$^{-1}$) enters the benthic food web. On the other hand, the carbon flows between bacteria and the environment declined by 7% (DOC→BAC), 34% (BAC→DOC), and 16 % (BAC→DIC) in GS1 compared to GC1. Instead, the metazoan contribution increased due to higher mean flows between the compartments of meiofauna and macrofauna. Likewise, in GS1, the meiofauna-related flows were one or two orders of magnitude higher than those related to macrofauna.

The biological system within the GC1 food web received around six times more OC flux than GS1. The primary distinction lay in the quantity of OC flux degraded from detritus to the DOC pool. The incoming OC flux in the biological system was roughly equivalent to the measured TOU in GC1 (Table 3) and approximately 24% of the measured TOU in GS1. The values of the internal flows (black flows in Fig. 4) representing the biological interactions in GS1 were higher than GC1,

corresponding to the higher biomass of meiofauna and macrofauna on the slope (Table 2). Since the bacteria stock

accounted for a larger proportion of the biomass in the canyon (~96%) compared to the slope (~31%, Table 2), the energy flow between bacteria and sediment stock was approximately 1.1 times (DOC→BAC) and 1.5 times (BAC→DOC) greater in GC1 than in GS1 (Fig. 4 and Table A7). On the other hand, the energy loss through predation and respiration was much lower in GC1 than in GS1, consistent with the significantly lower meiofauna and macrofauna standing stocks in the canyon head.

Within the benthic food webs, more than 99% of the prokaryotic production in GC1 is channeled to mortality and returns to the DOC pool (Fig. 5). In the GS1, 63.3% and 35.4% of the prokaryotic productions are directed toward mortality and meiofauna grazing, respectively. The relative fates of meiofaunal and macrofauna productions are similar between the two habitats. For meiofauna, the production is primarily transferred to mortality (i.e., ~32.6% to 33.3% returning to detritus), maintenance respiration (~20.2% to 23.1%), and megafauna predation (~31.8% to 36.7%). For macrofauna, the productions

mainly go to maintenance respirations (~72.6% to 73%).

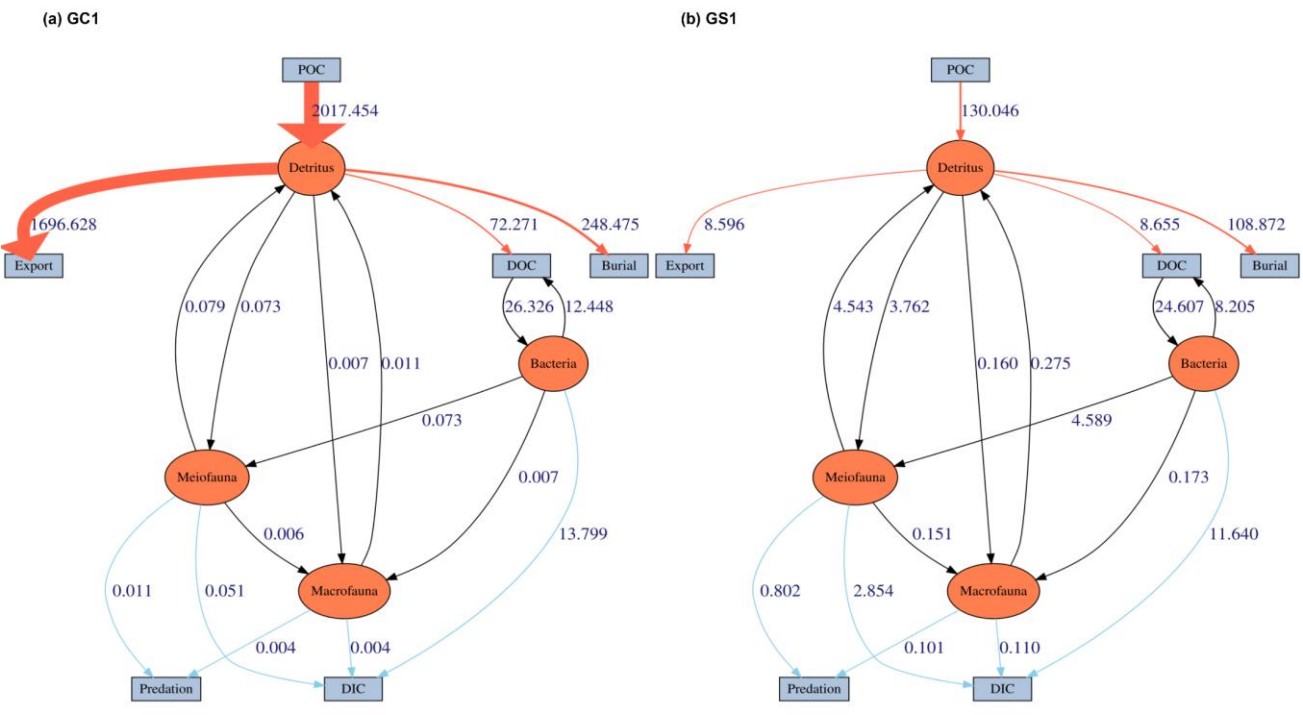

**Figure 4. The food web model derived from LIM results of (a) GC1 and (b) GS1 processed with the MCMC algorithm.**

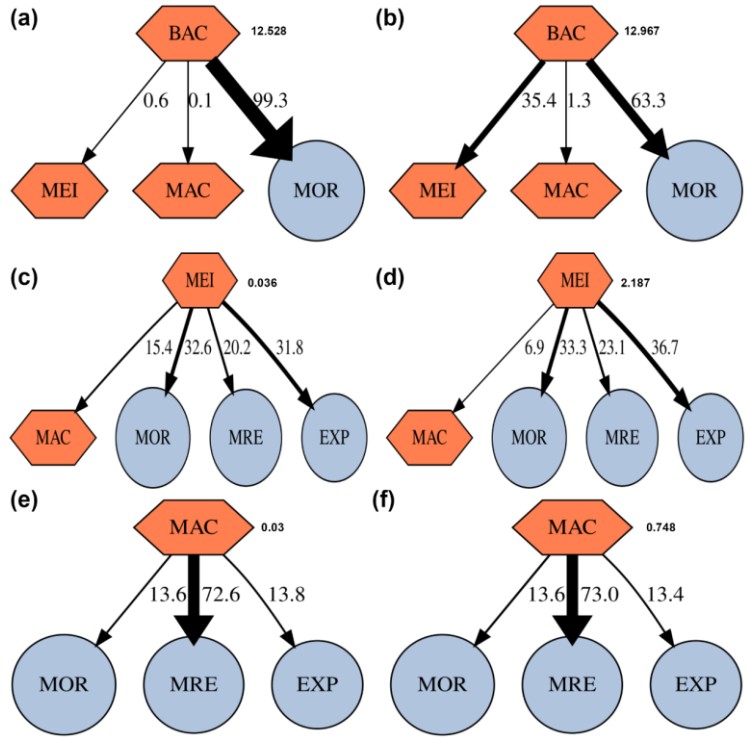

**Figure 5. Fate of secondary production (%) of prokaryotes, meiofauna, and macrofauna in GC1 (a, c, e) and GS1 (b, d, f).**
**Absolute production (mg C m$^{-2}$ d$^{-1}$) is indicated next to the compartment. The possible fates of this secondary production are**
**mortality (MOR), predation by meiofauna (MEI), macrofauna (MAC), maintenance respiration (MRE), and predation by**
**megafauna (EXP).**

### 3.6 Turnover time

Both measurement and modeling results showed that TOC turnover was much slower in GS1 (Table 3). The TOC turnover based on TOU was about 1.2–1.6 times slower in GS1 than in GC1, and the TOC turnover based on model results was about 3.5–5.0 times slower than that based on measured TOU for both sites. Bacteria turnover rates in two habitats calculated by different methods were comparable, with 4.6–4.7 days in GC1 and 3.7–4.8 days in GS1. However, the turnover time for meiofauna and macrofauna estimated by the two methods was distinctly different. In GC1, the turnover time estimated by the observation-based BMU was ~800 times shorter than those calculated by the model. Though the turnover time difference between the two methods in GS1 was not as vast as in GC1, it was still 14 times longer by model estimation. The turnover time was much shorter in GC1 with the measured BMU (GC1: 0.1 days; GS1: 2.3 days), while the turnover times estimated by models in GC1 (50.0 days) were about 1.5 times higher than in GS1 (32.5 days).

**Table 3. Oxygen utilization (mg C m$^{-2}$ d$^{-1}$) and stock turnover (unit: year or day) calculated from measured and modeled oxygen**
**utilization rates. Note that for the modeled results, TOU was defined as the sum of BAC→DIC, MEI→DIC and MAC→DIC,**
**DOU was defined as BAC→DIC, BMU was defined as the sum of MEI→DIC and MAC→DIC in Fig. B1.**

| | Source | GC1 | GS1 |
|---|---|---|---|
| Observed Oxygen Utilization | TOU | 69.6 ± 39.92 | 51.8 ± 15.23 |
| | DOU | 14.4 ± 15.78 | 9.0 ± 9.49 |
| | BMU | 49.0 ± 28.24 | 41.2 ± 12.57 |
| Modeled Oxygen Utilization | TOU | 13.9 ±5.59 | 14.6 ± 3.67 |
| | DOU | 13.8 ± 5.59 | 11.6 ± 3.22 |
| | BMU | 0.06 ± 0.01 | 2.96 ± 1.11 |
| Observed Turnover Time | $OC_{total}/TOU$ (yr) | 17.44 | 28.64 |
| | $OC_{BAC}/DOU$ (d) | 4.55 | 4.78 |
| | $OC_{MEI+MAC}/BMU$(d) | 0.06 | 2.34 |
| Modeled Turnover Time | $OC_{total}/TOU$ (yr) | 87.59 | 101.49 |
| | $OC_{BAC}/DOU$ (d) | 4.73 | 3.68 |
| | $OC_{MEI+MAC}/BMU$(d) | 49.95 | 32.54 |

**3.7 Network indices**

Total system throughput (T..) and total system throughflow (TST) were significantly higher in the food web of GC1 than in that of GS1 (Fig. 6). Total system cycled throughflow ($TST_C$) was not statistically different between the two sites, while the Finn cycling index (FCI) and average mutual information (AMI) of GC1 were significantly lower than those of GS1 (Fig. 6).

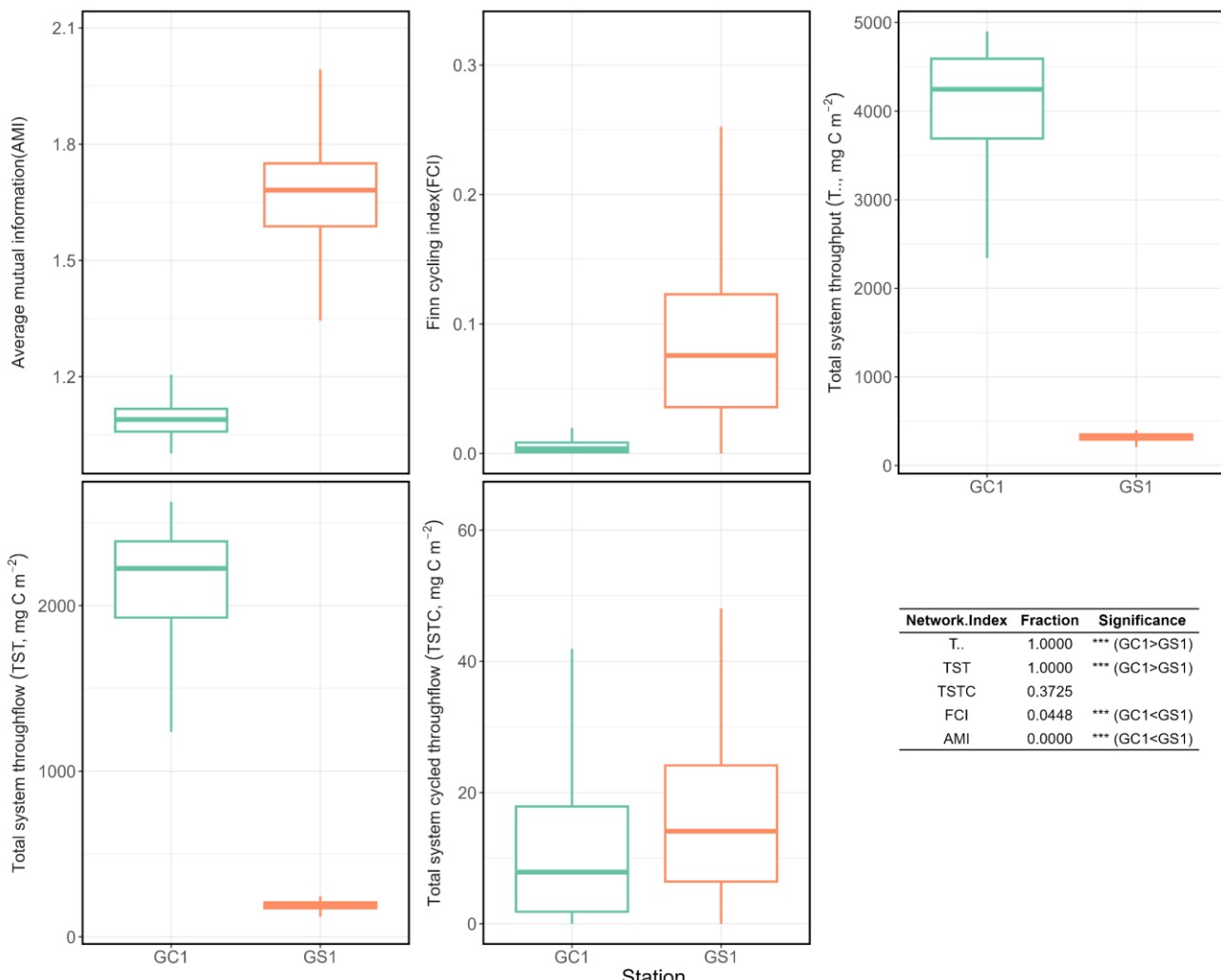

**Figure 6. Selected network indices of GC1 and GS1. The indices were calculated from 10,000 solutions of the LIM. As a result, there were also 10,000 values for each index, presented as box plots. The table at the bottom right compares network indices calculated for GC1 and GS1. The number indicates the fraction of network values higher in GC1 than in GS1.**

| Network.Index | Fraction | Significance |
|---|---|---|
| T.. | 1.0000 | *** (GC1>GS1) |
| TST | 1.0000 | *** (GC1>GS1) |
| TSTC | 0.3725 | |
| FCI | 0.0448 | *** (GC1<GS1) |
| AMI | 0.0000 | *** (GC1<GS1) |

## 4 Discussion

### 4.1 OC stocks and study limitations

GPR and GPSC exhibit distinct dry and wet seasons due to the influence of monsoons and typhoons, resulting in extreme precipitation in the region from May to October (Liu et al., 2016). The combination of steep topography, highly erodible drainage basin, and intense human activities also lead to remarkably high suspended-sediment load and fluvial discharges from the GPR during the wet season (Huh et al., 2009; Liu et al., 2013; 2016). However, the sediment and macrofauna OC stocks were not statistically different among sampling seasons (Fig. 2a, 2d). A possible explanation for the lack of seasonal variation in the OC stocks is the sampling timing. Our sampling cruises were mainly conducted in spring (March to April) and autumn (October to November) during the dry period, with only one summer sampling (August, Table A1). The OC stocks could have recovered from presumably more perturbed wet periods and, thus, without a notable seasonal difference.

The lack of seasonality prompts us to use the average stocks of spring and autumn in the LIM model, representing the food web dynamic during the dry period.

The sediment OC in the head of GPSC is lower than in the previous submarine canyon studies by Rowe et al. (2008) and van Ovelen et al. (2011). For instance, the TOC content in the Mississippi Canyon head is approximately 1.9% (Rowe et al., 2008), considerably higher than the average TOC content in the head of GPSC (0.39%). Bacteria biomass is the largest

living component of OC stock in the head of GPSC (Fig. 2b). Still, it is less than half of the value in the Mississippi Canyon head (2,585 mg C m$^{-2}$). Bacteria biomass and production can be affected by temperature, physical disturbance, substrate type, and sediment composition (Yamamoto and Lopez, 1985; Alongi, 1988). Seasonal bacterial population and biomass variations have also been observed from the shallow to deep Baltic (Meyer-Reil, 1983) and Cretan Seas (Danovaro et al., 2000); however, the controlling factor is site-specific (van Duyl and Kop, 1994). As the bacterial biomass was also low in

GS1, the low bacterial biomass may be a shared feature for the sediment of this SMR-fed active margin.

In food web studies, the general trophic structure of communities is established by grouping organisms into broader feeding types based on information gathered from the literature (e.g., Fauchald and Jumars 1979; Lincoln 1979; Dauwe et al. 1998). However, the lack of adequate taxonomic resolution (e.g., family, genus, or species level) hinders our ability to segregate data into functional groups. Consequently, we derived estimates of the metazoan carbon flows solely based on

physiological constraints and mass balancing rather than relying on the feeding preferences of the functional groups. Moreover, some metazoans can switch their feeding behavior in response to changing food quality and quantity in the environment. For instance, some Polychaeta and Mollusca species can transition from deposit-feeding to suspension-feeding or vice versa following fluctuations in the flux of suspended particles (Taghon and Greene, 1992), providing an advantage in environments with varying flow velocities. A mesocosm observational study also demonstrated that two

surface deposit feeder species, *Sipunculus* sp. (Sipuncula) and *Spiophanes kroeyeri* (Polychaeta), shifting from deposit-feeding to suspension-feeding under different flow conditions (Thomsen and Flach, 1997). Thus, while aggregating all metazoans into two size classes oversimplifies the food web model, it may help prevent misrepresenting the relative contributions of taxa exhibiting high feeding plasticity.

On the other hand, the organic matter degradation in sediments exhibits high variability, influenced by factors such as

organic matter chemistry, sediment physical characteristics, and biological agents involved in decomposition (Middelburg and Meysman, 2007). As a result, the proportions of detrital organic carbon components (i.e., labile, semi-labile, and refractory fractions) typically differ across study locations and cannot be directly inferred from previous literature (e.g., van Oevelen et al. 2011; Dunlop et al. 2016; Stratmann et al. 2018; Durden et al. 2020). Due to lacking empirical data, we aggregated all detritus into a single compartment, consistent with the approach in other benthic food web model studies

(e.g., Rowe et al. 2008). Additionally, organic matter degradation occurs over a broad spectrum of time scales, from minutes for biochemical breakdown in animal guts to $10^6$ years for organic carbon mineralization in deep-sea sediments (Middelburg et al., 1993). This wide dynamic range underscores the significant influence of the characteristic time scale of experimental observations on measurable degradation rates (Hedges and Keil, 1995). Given that our model measures sediment organic carbon oxidation through oxygen consumption, it likely emphasizes the estimation of labile organic

carbon degradation over semi-labile and refractory fractions due to their extended time scales.

The current food web models omit megafauna stock due to data limitations, notably the technical challenges of trawling in hazardous areas like the GPSC. Consequently, the megafauna predation is only constrained by the net growth efficiency of

meiofauna and macrofauna, with all metazoan growths assumed to be consumed by megafauna. In the deep sea, the megafauna abundance and biomass are generally lower and decrease more rapidly with depth than smaller infauna in the deep sea (Rex et al., 2006; Wei et al., 2010). However, certain areas can exhibit high megafauna and fish densities (Sibuet, 1977; Hecker, 1994; Fodrie et al., 2009), which can influence the redistribution and quality of OM in the marine sediments (Smallwood et al., 1999), potentially affecting the food web dynamics. While van Oevelen et al. (2011) highlighted the significance of deposit-feeding megabenthos at the mid-section of the Nazaré Canyon, their sampling depths are considerably deeper than ours. It is suggested that the intense flow regimes at the canyon head (i.e., tidal and turbidity currents) may favor the mobile megafauna (i.e., fish, crabs, and octopus) over the deposit-feeding counterparts (i.e., sea urchins) because the mobile megafauna predators are better coping with physical disturbance and may benefit from transient food subsidy in the canyon (Vetter and Dayton, 1999; Vetter et al., 2010). Per the presented model structure, the sediment detrital carbon in GPSC would be directed toward export, burial, microbial loop, and smaller metazoan. Nonetheless, the absence of deposit-feeding megafauna is not expected to disproportionately influence the GPSC food web.

## 4.2 Implemented model constraints

LIM food webs combine several physiological and geochemical constraints as lower and upper bounds of the carbon flows. This creates the possible solution space for the algorithm to iterate and find a solution set of all flows. However, most LIM studies used reference constraints due to technical difficulties in physiological experiments on benthic species (van Oevelen et al., 2010). For example, Stratmann et al. (2018) studied the abyssal plain food web of the Peru Basin; however, some physiological constraints of benthos in the food webs were derived from the shallow water or intertidal species (e.g., Drazen et al., 2007; Koopmans et al., 2010). In other food web studies, the physiological constraint of the dominant species was used as the representative for the entire size group (e.g., De Smet et al., 2016). Though these low-quality constraints bring some uncertainty to our model results, the flow solution was still more convincing than those without constraints (van Oevelen et al., 2010).

In the LIM food webs, the maximum bacterial OC remineralization (BAC→DIC) was set to 30% of the measured TOU (cf. Section 2.4.4). In fact, the measured DOUs were 21% of the measured TOU in GC1 and 17% of the measured TOU in GS1, suggesting that the maximal constraints were reasonable. Furthermore, the modeled DOU flows (BAC→DIC) agreed well with the direct measurement in both study sites, and the modeled TOU flows (sum of BAC→DIC, MEI→DIC, and MAC→DIC) were within the same order of magnitude as the measured TOU (Table 3), lending credibility to the LIM results.

The main difficulty lies in modeling BMU, which includes faunal respiration and the oxygen uptake related to biological activities (Glud et al., 2003). Macrobenthos may modify diagenetic reactions, sediment–water exchange, and sediment composition (Aller, 1982, 1994) through their activities, such as feeding, burrowing, tube construction, bioirrigation (Jørgensen et al., 2005), and bioresuspension (Graf and Rosenberg, 1997). Forster and Graf (1995) reported that two macrofauna species, *Callianassa subterranea* (Decapoda) and *Lanice conchilega* (Polychaeta), enhance TOU by 85% through their pumping behavior in the shallow North Sea. The animal-induced changes in oxygen distribution are notoriously difficult to quantify and separate from their maintenance respiration. Therefore, the most straightforward and robust procedure to evaluate the fauna activities in mixed communities is to subtract DOU from TOU (Glud, 2008). Unfortunately, the current food web structure considers only the respiration of meiofauna and macrofauna, which is far

less than the observed BMU (Table 3). The total biomass of meiobenthos and macrobenthos was significantly higher on the slope, which explained the higher modeled BMU in the GS1. However, the observed BMU was not statistically different between the two sites, suggesting that the biomass may not predict bioturbation and, thus, the observed BMU. Liao et al. (2017) found that several polychaete families capable of burrowing deep into the sediment, including paraonids, cossurids, capitellids, and sternaspids, thrived in the canyon habitats. In contrast, the discretely motile, surface deposit, and suspension

feeders (e.g., cirratulids, ampharetids, and spionids) dominated the slope sediment but diminished in the canyon (Liao et al., 2017). Therefore, the discrepancy between the modeled and observed BMU in GC1 may be related to the active bioturbation behaviors of these subsurface deposit feeders.

TOU is the most reliable proxy of total benthic carbon degradation in marine sediments because it integrates aerobic activity, nitrification, and re-oxidation of reduced inorganic compounds (Stratmann et al., 2019). Therefore, the TOU should reflect

the variation in POC input. However, POC flux determined by sediment traps usually does not match the carbon demand of the benthic community (Smith, 1987; Jørgensen et al., 2022). This study used reported vertical POC flux data as constraints to approximate OC input in the food webs. While our modeled POC flux is approximately 29 times higher than the measured TOU in the canyon head, the OC flux supporting the GC1 biological system (72.4 mg C $m^{-2}$ $d^{-1}$) agreed well with the measured TOU (69.6 mg C $m^{-2}$ $d^{-1}$). By contrast, the modeled POC flux is about two times higher than the measured

TOU on the nearby slope; however, it only meets 19% of the benthic OC demands (i.e., measured TOU). Similarly, Smith and Kaufmann (1999) and Smith et al. (2016) reported a long-term discrepancy between food supply and demand in the Eastern North Pacific, with the POC fluxes contributing only around half of the TOU. Moreover, the imbalance between carbon supply and demand varied by region on the abyssal plain in the deep Arabian Sea (Witte and Pfannkuche, 2000), in which the sediment trap fluxes matched 50% of the benthic carbon demand in the southernmost station but only 20% in

the westernmost station. These mismatches were mainly explained as the uncertainties concerning relatively short-term TOU incubations via long-term sediment trap deployment (Witte and Pfannkuche, 2000). The sediment traps also fail to capture episodic flux events due to their low temporal resolution (Huffard et al., 2020). Besides, the benthic oxygen consumption rates can be affected by primary production, quality of OM, and bottom-water oxygen concentrations (Jahnke, 1996; Wenzhöfer and Glud, 2002). It has been reported that the dissolved oxygen concentration in the bottom water

increased toward the canyon head of GPSC (Liao et al., 2017). Also, Wang et al. (2008) reported that the flow velocity near the head of GPSC regularly exceeded 1 m $s^{-1}$. The high flow velocity may enhance water mass mixing and thus increase oxygen concentration at the canyon head. In short, the peculiarly high value of observed TOU at the canyon head might be associated with higher bacteria stocks and remineralization, underestimated bioturbation, physical disturbances, or chemical oxidation.

**4.3 Network characteristics**

Network indices are often calculated to describe the function of the food web (Kones et al., 2009). Here, five indices were used to examine the food web structures in GC1 and GS1. TST and T.. quantify the energy that belongs to the system as the sum of flows. The magnitude of these two indices is directly linked to the ecosystem's growth. The greater TST and T.. in the canyon head (Fig. 6) implies more intense carbon processing and a higher total amount of energy flowing through

the system (Ulanowicz, 1986; 2004). Notably, the microbial loop (de Jonge et al., 2020), encompassing detritus degradation, DOC assimilation by prokaryotes, viral-induced prokaryote mortality, DOU, and metazoan predation on prokaryotes,

reveals a considerable contribution to T... in the two systems. Specifically, the microbial loop constitutes 3% of T... in GC1 (124.9 mg C m$^{-2}$ d$^{-1}$) and 18% of T... in GS1 (57.9 mg C m$^{-2}$ d$^{-1}$). The higher input to the DOC pool in the canyon supports a greater prokaryotic biomass.

However, TST and T.. cannot provide details about how the material is distributed within the system. In other words, two systems with totally different food web structures can have the same TST (Bodini, 2012). Therefore, FCI and AMI were computed to estimate the maturity of the two ecosystems. According to Odum (1969), a mature ecosystem (i.e., with higher FCI values) should involve a higher information content, high biomass, and a high capacity to seize and hold the nutrients for cycling within the system. Our finding shows that FCI is significantly greater in GS1 than in GC1, indicating that the

slope system has a higher degree of energy recycling and is a relatively mature ecosystem that develops more completed nutrient conservation routes (Odum, 1969) compared to the canyon system. Moreover, the reported FCI of benthic ecosystems, including one submarine canyon, ranges between 5% and 20% (van Oevelen et al., 2011; Kones et al., 2009; Anh et al., 2015). While the FCI of GS1 (7.6%) falls within this range, that of GC1 is much lower (0.4%), resulting from the relatively large value of TST. These results also indicated that cycling had a modest impact on the TST in the GC1

model. Though the cycle index alone cannot assess the ability to resist environmental change in a system, perturbations can substantially influence the cycling process of an ecosystem (Saint-Béat et al., 2015). The higher biodiversity (Liao et al., 2017, 2020) and biotic carbon stock suggest that GS1 is a more mature ecosystem than the frequently disturbed GC1. In contrast, GC1 is dominated by the bacteria-related process (e.g., higher bacteria stocks and OC flows) with the lower metazoan carbon stock and the associated flow, as demonstrated by the modeled fates of secondary production from this

(Fig. 5) and other studies (Oevelen et al., 2011). For instance, despite the increased OC flow to the DOC pool and higher prokaryotic biomass in the GPSC, the energy transfer to higher trophic levels is hindered by the lack of metazoans, leading to the predominant fate of prokaryotic production toward mortality (99.3%). On the slope, while prokaryotic mortality remains the primary fate (63.3%), considerable production transfers to higher trophic levels through meiofauna grazing (35.4%).

Ulanowicz (1980) proposed that AMI is a better indicator of an ecological network's developmental status to measure the maturity of an ecosystem. Since the energy distribution and pathway determine the maintenance cost for the whole system, Bodini (2012) suggested that a highly redundant network that is less organized would have a lower AMI value. Also, trophic specialization is expected to result in higher values for AMI (Ulanowicz, 2004). Conversely, low AMI value reflects the early immature stage of the ecosystem (Mukherjee et al., 2019). As expected, the highly disturbed canyon head has a

significantly lower AMI than the upper slope.

Turnover time derived from the direct measurements of TOU, DOU, or BMU was much shorter than that derived from modeled estimations. The longer OC turnover rates estimated by models have resulted from an underestimation of the bioturbation or physical disturbance in the natural environment. The turnover time of total OC estimated by direct measurement in both sites was in the time scale of decades, consistent with the previous study by Rowe et al. (2008) in the

Mississippi Canyon. Besides, the longer OC residence time in GS1 corresponds to the interpretation of FCI, revealing that more OC is recycled in the slope habitat.

## 4.4 Fate of OC in the source-to-sink system

The fate of OC in source-to-sink systems is a topic of everlasting interest and discussion within the biogeoscience community. From the perspective of system budget, earlier studies estimated a burial efficiency, defined as the proportion of buried flux relative to riverine POC load, of 11–13% in the Gaoping shelf–slope–canyon system (Hsu et al., 2014; Hung et al., 2012). Our model-based burial efficiency of 12% in the GC1 model (Table A8) is consistent with these estimates, but the GS1 model yields a much higher value (84%). This mismatch can be attributed to earlier studies not distinguishing the canyon from the slope in their estimates. Given the distinct hydrodynamic conditions within the canyon and on the slope (Liu et al., 2002; Liu et al., 2016), our site-specific estimates adequately capture the heterogeneous nature of burial efficiency at this active margin.

Other studies approached the issue by comparing OC's isotopic and elemental composition in buried sediments to those of riverine POC (e.g., Galy et al., 2007). This approach concluded that due to frequent hyperpycnal flows and rapid deposition on the narrow continental margin, OC in SMR-fed continental margin sediment typically exhibits high fidelity to riverine POC, ranging from 70–85% (Blair and Aller, 2012). The GPR-sustained margin, a typical SMR-sustained carbon source-to-sink system, has a similar feature: Kao et al. (2014) found that the OC fidelity was ~100% in the GPSC sediment, but dropped to ~70% on the slope sediment. In our model, OC fidelity can be approximated as the proportion of buried flux relative to the flux of OC initially retained on the seabed (Table A8). Our modeling results show that the fidelity is 77% in GC1 and 90% in GS1, which agrees with the geochemistry-based approach by Kao et al. (2014). Nevertheless, our result should be viewed as an overestimate of the fidelity because of the underestimated modeled BMU (Table 3). If the unaccounted TOU (i.e., the difference between observed and modeled TOU) were added to the OC flux retained on the seabed, the fidelity would drop to 66% in GC1 and 69% in GS1 (Table A8).

The extraordinary OC export rate in the canyon head may be attributed to sediment resuspension or lateral transports due to strong internal tide energy, as observed by Liu et al. (2006; 2009). Considering that the estimated OC export rate in the canyon head is almost eight times greater than on the adjacent upper slope (Table A8), our finding suggests that while the GPSC may be a short-term sink within a season, it functions primarily as a conduit for sediment transport to deep ocean basins in the long run (Liu et al., 2006; Huh et al., 2009; Yu et al., 2009; Kao et al., 2010; Chiou et al., 2011). By contrast, sediment deposition across the upper GS is influenced by the "filling-and-spilling" process (Hsu et al., 2013), with competition between local basin flank uplift and sediment deposition rates leading to higher accumulation rates.

As the cycling and sequestration of OC in seafloor sediments are crucial in deep-sea ecosystems (Dunne et al. 2007; Thurber et al. 2014), understanding the fate of POC is essential. Our study investigates the amount of OC entering the biological system, which was not previously determined through geochemical studies. Surprisingly, the energy supporting a highly disturbed canyon food web is disproportionately low (4%, Table A8) compared to the significant influx of POC. In contrast, the adjacent slope receives approximately an equal amount of OC flux into its biological system, despite a significantly lower influx of POC. The strong physical disturbances in GC1 have negatively impacted the benthic community structure (Tung et al., 2023), resulting in meager meiofauna and macrofauna biomass and interaction flows. Nevertheless, the higher bacteria stocks and bacteria-related carbon flows could compensate for these deficiencies. Our models show the inner workings of carbon cycling on the seabed with the advantage of disentangling the OC entering the food web, contributing to long-term burial, and exiting the system.

Having reconstructed the sedimentary OC cycling during dry periods in both the canyon and slope, the reasonable next step involves developing models that establish connections to driving factors of the system, such as variations in river water flux or sediment load. These endeavors necessitate a collaborative approach, combining high-frequency data collection with more advanced model design techniques. The ultimate objective is to enhance our capacity to predict the response of benthic communities and OC cycling under the influence of climate change. This is especially pertinent given the anticipated increase in flooding intensity and frequency, which may lead to heightened disturbances in submarine canyons (Canals et al., 2006; Chao et al., 2016; Huang and Swain, 2022; Sequeiros et al., 2019).

## 5 Conclusions

This study provides the first quantitative analysis of carbon flows within the benthic food webs of GPSC and the adjacent slope. The amount of OC stocks differed significantly between the canyon and slope but not among sampling seasons. Nevertheless, there was no significant difference in oxygen utilization, including TOU, DOU, and BMU, between habitats or among the sampling seasons. Therefore, LIM food web models were applied to the average OC stocks during dry periods, assuming mass balance with geochemical and physiological constraints on energy flows, to gain insight into ecosystem functioning and food web characteristics. Using network indices, including total system throughput, energy recycling, and food web maturity, we found more OC flowing through the canyon head than the upper slope. Bacteria-related processes dominated the canyon head food web. In contrast, meiofauna and macrofauna contributed more to carbon processing on the adjacent slope, implying a relatively mature ecosystem and significantly higher energy cycling. Furthermore, the food web models accurately predict oxygen utilization, demonstrating a good match between modeled and observed oxygen utilization. The turnover time, calculated from the ratio between OC stocks and oxygen utilization, was much longer on the slope than in the canyon, corresponding to a higher degree of energy recycling indicated by network indices. The low faunal carbon stock in the canyon head suggests that it may be a fragile ecosystem under severe physical perturbation. Despite the high OC input, only 4% of the OC enters the canyon food web, while the remaining OC either exits the system or buries in the sediment. This study confirms the significant role of GPSC in transporting OC to the deep South China Sea, contributing to our understanding of ecosystem functioning in the area and improving our knowledge of natural carbon transfer within the "Source-to-Sink" system.

**6 Appendice A**

**Table A1. Sampling cruises including dates, coordinates, and water depths of two sites. Different gears were used to collect sediment and benthic fauna samples. Cruises were assigned to the corresponding seasons to test the seasonal effect.**

| Date | Season | Cruise | Station | Longitude (°E) | Latitude (°N) | Depth (m) | Measurement | Gear |
|---|---|---|---|---|---|---|---|---|
| Jan 2014 | AU | OR1_1096 | GC1 | 120.4170 | 22.4170 | 222 | Sediment TOC, Macrofauna | CTD, Boxcorer |
| | | | GS1 | 120.4006 | 22.2349 | 270 | Sediment TOC, Macrofauna | CTD, Boxcorer |
| Mar 2015 | SP | OR1_1099 | GC1 | 120.3768 | 22.4017 | 395 | Sediment TOC, Macrofauna | CTD, Multicorer |
| | | | GS1 | 120.4002 | 22.2328 | 277 | Sediment TOC, Macrofauna | CTD, Multicorer |
| Apr 2015 | SP | OR1_1102 | GC1 | 120.4114 | 22.4173 | 323 | Sediment TOC, Macrofauna | CTD, Boxcorer |
| | | | GS1 | 120.4006 | 22.2329 | 279 | Sediment TOC, Macrofauna | CTD, Boxcorer |
| Aug 2015 | SU | OR1_1114 | GC1 | 120.4114 | 22.4172 | 320 | Sediment TOC, Meiofauna, Macrofauna | CTD, Multicorer |
| | | | GS1 | 120.3998 | 22.2322 | 279 | Sediment TOC, Meiofauna, Macrofauna | CTD, Multicorer |
| Nov 2015 | AU | OR1_1126 | GC1 | 120.4112 | 22.4175 | 318 | Sediment TOC, Meiofauna, Macrofauna, DOU | CTD, Multicorer |
| | | | GS1 | 120.3995 | 22.2329 | 277 | Sediment TOC, Meiofauna, Macrofauna, DOU | CTD, Multicorer |
| Feb 2016 | SP | OR1_1128 | GC1 | 120.4108 | 22.4171 | 319 | Sediment TOC, Meiofauna, Macrofauna, TOU, DOU | CTD, Multicorer |
| Mar 2016 | SP | OR1_1132 | GS1 | 120.4008 | 22.2296 | 281 | Meiofauna, Macrofauna | CTD, Multicorer |
| Oct 2016 | AU | OR1_1151 | GC1 | 120.4110 | 22.4176 | 317 | Sediment TOC, Macrofauna, TOU, DOU | CTD, Multicorer |
| | | | GS1 | 120.4004 | 22.2315 | 272 | Sediment TOC, Macrofauna, DOU | CTD, Multicorer |

| Date | Season | Cruise | Station | Longitude (°E) | Latitude (°N) | Depth (m) | Measurement | Gear |
|---|---|---|---|---|---|---|---|---|
| Mar–Apr 2018 | SP | OR1_1190 | GC1 | 120.4104 | 22.4172 | 321 | Sediment TOC, Bacteria, Macrofauna, TOU, DOU | CTD, Multicorer |
| | | | GS1 | 120.3977 | 22.2309 | 279 | Sediment TOC, Bacteria, Macrofauna, TOU, DOU | CTD, Multicorer |
| Mar–Apr 2019 | SP | OR1_1219 | GC1 | 120.4092 | 22.4179 | 318 | Sediment TOC, TOU, DOU | CTD, Multicorer |
| | | | GS1 | 120.3772 | 22.2538 | 246 | Sediment TOC, TOU, DOU | CTD, Multicorer |
| Oct 2019 | AU | OR1_1242 | GC1 | 120.4082 | 22.4168 | 271 | Sediment TOC, TOU, DOU | CTD, Multicorer |
| | | | GS1 | 120.3776 | 22.2531 | 261 | Sediment TOC, TOU, DOU | CTD, Multicorer |
| Nov 2020 | AU | NOR1_T004 | GC1 | 120.4121 | 22.4140 | 309 | Sediment TOC, TOU, DOU | CTD, Multicorer |

**Table A2 Reference of geochemical data and derivation of geochemical constraints in Table 1.**

| Parameter | Calculation | GC1 | GS1 | Unit | Reference |
|---|---|---|---|---|---|
| Bulk density | | 2.1 | 1.8 | g cm$^{-3}$ | Su et al. (2018) |
| Water content | | 0.4 ± 0.13 | 0.4 ± 0.06 | - | |
| Dry density | $(1 - \text{water content}) \times \text{bulk density}$ | 1.3 ± 0.28 | 1.1 ± 0.11 | g cm$^{-3}$ | This study |
| TOC content | | 0.4 ± 0.11 | 0.5 ± 0.04 | - | |
| SAR | | - | 0.7 | g cm$^{-2}$ y$^{-1}$ | Huh et al. (2009) |
| | | 0.723 | - | cm y$^{-1}$ | Tsai and Chung (1989) |
| Burial rate | SAR × TOC content | 162.34 | 100.92 | mg C m$^{-2}$ d$^{-1}$ | This study |
| Measured mass flux | | [200, 400] | - | g m$^{-2}$ d$^{-1}$ | Liu et al. (2006); Liu et al. (2009) |
| Measured POC flux | | - | [42, 115] | mg C m$^{-2}$ d$^{-1}$ | Shih et al (2020) |
| Trapping efficiency | | - | 0.8 | - | Hung and Gong, 2007 |
| | | - | 0.75 | - | Li, 2009 |
| POC flux constraints | Measured mass flux × TOC content[a] | [800, 2400] | - | mg C m$^{-2}$ d$^{-1}$ | This study |
| | Measured POC flux/Trapping efficiency | - | [52.5, 153.3] | mg C m$^{-2}$ d$^{-1}$ | This study |
| Burial efficiency constraints | Burial rate/POC flux constraints | [0.067, 0.203] | [0.66, 1] | - | This study |
| Export rate constraints | | [0.82, 0.87] | [0, 0.87] | - | Huh et al. (2009) |

[a]Here TOC content was substituted with 0.4-0.6 % as reported in Liu et al. (2009)

**Table A3. Symbol nomenclature for calculating network (Kones et al., 2009). Assuming a system comprises *n*
compartments, the flow value *T_ij* is defined as a sink-to-source flow (i.e., *j→i*) following the notation by Latham
(2006). The compartments importing to the internal network are denoted by 0, the destination of usable exports
(secondary production) is labeled as n + 1, and the destination of unusable exports (respiration/dissipation) is
labeled as *n + 2* (Hirata and Ulanowicz, 1984).**

| Term | Description |
|---|---|
| n | Number of internal compartments in the network, excluding 0 (zero), $n+1$ and $n+2$ |
| $j = 0$ | External source |
| $j = n+1$ | Useable export from the food web (export) |
| $j = n+2$ | Unusable export from the food web (dissipation) |
| $T_{ij}$ | Flow from compartment $j$ to $i$ where $j$ represents the columns of the flow matrix and $i$ the rows |
| $T_{ij}^*$ | Flow matrix, excluding flows to and from the externals |
| $T_{i.}$ | Total inflows to compartment $i$ |
| $T_{.j}$ | Total outflows from compartment $j$ |
| $T_i$ | Total inflows to compartment $i$ excluding inflow from external sources |
| $T_j$ | Total outflows from compartment $j$ excluding outflow to external sources |
| $(x_i)_-$ | A negative state derivative, considered as a gain to the system pool of mobile energy |
| $(x_i)_+$ | A positive state derivative, considered as a loss from the system pool of mobile energy |
| $Z_{i0}$ | Flow into compartment $i$ from outside the network |
| $Y_{n+j}$ | Flow out of the network for compartment $j$ to compartments $n+1$ and $n+2$ respectively |
| $C_{ij}$ | The number of species with which both $i$ and $j$ interact divided by the number of species with which either $i$ or $j$ interact |
| $I$ | Identity matrix |

**Table A4. Formula of the network indices; see Table A3. for the symbols.**

| Index type | Index name | Symbol | Formula | Reference |
|---|---|---|---|---|
| General indices | Total system throughput | T.. | $$\sum_{i=1}^{n+2}\sum_{j=0}^{n} T_{ij}$$ | Hirata and Ulanowicz (1984) |
| General indices | Total system throughflow | TST | $$\sum_{i=1}^{n}\sum_{j=1}^{n}[\,T_{ij} + z_{i0} - (\dot{x_\iota})_-\,] = \sum_{i=1}^{n}\sum_{j=1}^{n}[\,T_{ij} + y_{n+j} - (\dot{x_\iota})_+\,]$$ | Latham (2006) |
| General indices | Total system cycled throughflow | TST$_C$ | $$\sum_{j=0}^{n}\left(1 - \frac{1}{q_{ij}}\right)T_j\,,\ \text{where } Q = (I - G')^{-1}, \text{where } G'$$ $$= [T_{ij}^*/\max(Ti, Tj)]$$ | Finn (1976; 1978; 1980), Patten and Higashi (1984); Patten et al. (1976) |
| Pathway analysis | Finn's cycling index | FCI | $$\frac{TST_C}{TST}$$ | |
| Network uncertainty | Average mutual information | AMI | $$k\sum_{i=1}^{n+2}\sum_{j=0}^{n}\frac{T_{ij}}{T..}\log_2\frac{T_{ij}T..}{T_{i.}T_{.j}}$$ | Ulanowicz (2004) |

 **Table A5. PERMDISP on carbon standing stocks and oxygen utilization of GC1 and GS1.**

|  | Source | Df | Sum of squares | Mean squares | F-value | Pr (>F) |
|---|---|---|---|---|---|---|
| Sediment | Habitat | 1 | 1.4936e+09 | 1493571093 | 0.6217 | 0.4469 |
|  | Residuals | 16 | 3.8441e+10 | 2402553001 |  |  |
|  | Season | 1 | 3.0389e+07 | 30388504 | 0.0061 | 0.9375 |
|  | Residuals | 16 | 7.9247e+10 | 4952928845 |  |  |
| Bacteria | Habitat | 1 | 101.79 | 101.786 | 2.363 | 0.1397 |
|  | Residual | 18 | 775.35 | 43.075 |  |  |
| Meiofauna | Habitat | 1 | 1669.30 | 1669.30 | 30.284 | **0.0001** |
|  | Residual | 16 | 881.94 | 55.12 |  |  |
|  | Season | 2 | 2339.55 | 1169.78 | 25.979 | **0.0001** |
|  | Residuals | 15 | 675.42 | 45.03 |  |  |
| Macrofauna | Habitat | 1 | 22338 | 22337.6 | 59.352 | **0.0001** |
|  | Residual | 43 | 16183 | 376.4 |  |  |
|  | Season | 1 | 2246 | 2245.7 | 0.9943 | 0.3136 |
|  | Residuals | 43 | 97117 | 2258.5 |  |  |
| TOU | Habitat | 1 | 1799.1 | 1799.10 | 3.3882 | 0.0763 |
|  | Residual | 25 | 13274.7 | 530.99 |  |  |
|  | Season | 1 | 194.4 | 194.36 | 0.3012 | 0.5893 |
|  | Residuals | 25 | 16133.2 | 645.33 |  |  |
| DOU | Habitat | 1 | 63 | 63.016 | 0.4053 | 0.5425 |
|  | Residual | 65 | 10106 | 155.469 |  |  |
|  | Season | 1 | 22.7 | 22.734 | 0.1453 | 0.7412 |
|  | Residuals | 65 | 10170.1 | 156.463 |  |  |
| BMU | Habitat | 1 | 390.5 | 390.47 | 1.2647 | 0.2786 |
|  | Residual | 18 | 5557.3 | 308.74 |  |  |
|  | Season | 1 | 107.0 | 107.02 | 0.3269 | 0.5699 |
|  | Residuals | 18 | 5893.7 | 327.43 |  |  |

**Table A6. PERMANOVA on carbon standing stocks and oxygen utilization of GC1 and GS1.**

|  | Source | Df | Sum of squares | R2 | F | Pr (>F) |
|---|---|---|---|---|---|---|
| Sediment | Season | 1 | 5.1228e+08 | 0.00296 | 0.0616 | 0.8077 |
|  | Habitat | 1 | 4.2745e+10 | 0.24670 | 5.1427 | **0.0399** |
|  | Season:Habitat | 1 | 1.3645e+10 | 0.07875 | 1.6416 | 0.2225 |
|  | Residual | 14 | 1.1637e+11 | 0.67159 |  |  |
|  | Total | 17 | 1.7327e+11 | 1.00000 |  |  |
| Bacteria | Habitat | 1 | 2532.6 | 0.57508 | 24.36 | **0.0005** |
|  | Residual | 18 | 1871.3 | 0.42492 |  |  |
|  | Total | 19 | 4403.9 | 1.00000 |  |  |
| Meiofauna | Season | 2 | 1870.2 | 0.20383 | 29.965 | **0.0006** |
|  | Habitat | 1 | 4579.3 | 0.49908 | 146.741 | **0.0001** |
|  | Season:Habitat | 1 | 2351.5 | 0.25628 | 37.677 | **0.0002** |
|  | Residual | 12 | 374.5 | 0.04081 |  |  |
|  | Total | 17 | 9175.5 | 1.00000 |  |  |
| Macrofauna | Season | 1 | 2817 | 0.02440 | 1.7618 | 0.1907 |
|  | Habitat | 1 | 45390 | 0.39317 | 28.3861 | **0.0001** |
|  | Season:Habitat | 1 | 1678 | 0.01454 | 1.0494 | 0.3094 |
|  | Residual | 41 | 65560 | 0.56789 |  |  |
|  | Total | 44 | 115446 | 1.00000 |  |  |
| TOU | Season | 1 | 20 | 0.00059 | 0.0143 | 0.9095 |
|  | Habitat | 1 | 1657 | 0.04973 | 1.2060 | 0.2867 |
|  | Season:Habitat | 1 | 41 | 0.00122 | 0.0295 | 0.8687 |
|  | Residual | 23 | 31598 | 0.94846 |  |  |
|  | Total | 26 | 33315 | 1.00000 |  |  |
| DOU | Season | 1 | 69.2 | 0.00515 | 0.3355 | 0.5836 |
|  | Habitat | 1 | 371.7 | 0.02769 | 1.8035 | 0.1925 |
|  | Season:Habitat | 1 | 0.0 | 0.00000 | 0.0001 | 0.9935 |
|  | Residual | 63 | 12984.9 | 0.96716 |  |  |
|  | Total | 66 | 13425.8 | 1.00000 |  |  |
| BMU | Season | 1 | 16.6 | 0.00154 | 0.0253 | 0.8666 |
|  | Habitat | 1 | 274.0 | 0.02538 | 0.4174 | 0.5431 |
|  | Season:Habitat | 1 | 0.4 | 0.00004 | 0.0005 | 0.9817 |
|  | Residual | 16 | 10505.6 | 0.97304 |  |  |
|  | Total | 19 | 10796.6 | 1.00000 |  |  |

**Table A7. Mean and standard deviation of the food web flows (mg C m$^{-2}$ d$^{-1}$) of the GC1 and GS1 processed with the MCMC algorithm.**

| Flow | GC1 | | GS1 | |
|---|---|---|---|---|
| | Mean | St. dev | Mean | St. dev |
| POC_W→DET | 2017.454 | 308.2665 | 130.046 | 20.4974 |
| DET→BUR | 248.475 | 80.0207 | 108.872 | 24.6910 |
| DET→EXP | 1696.628 | 260.6000 | 8.596 | 6.7978 |
| DET→DOC | 72.271 | 48.4903 | 8.655 | 7.0644 |
| DOC→BAC | 26.326 | 13.6431 | 24.607 | 8.3267 |
| BAC→DOC | 12.448 | 9.2007 | 8.205 | 5.7025 |
| BAC→MEI | 0.073 | 0.0466 | 4.589 | 3.7226 |
| BAC→MAC | 0.007 | 0.0050 | 0.173 | 0.1254 |
| BAC→DIC_W | 13.799 | 5.5871 | 11.640 | 3.2161 |
| DET→MEI | 0.073 | 0.0471 | 3.762 | 2.5722 |
| MEI→MAC | 0.006 | 0.0043 | 0.151 | 0.1151 |
| MEI→DIC_W | 0.051 | 0.0134 | 2.854 | 1.1077 |
| MEI→PRE | 0.011 | 0.0075 | 0.802 | 0.4966 |
| MEI→MEI_DEF [a] | 0.067 | 0.0224 | 3.816 | 1.8648 |
| MEI→MEI_MOR [a] | 0.012 | 0.0075 | 0.727 | 0.4762 |
| DET→MAC | 0.007 | 0.0050 | 0.160 | 0.1196 |
| MAC→DIC_W | 0.004 | 0.0010 | 0.110 | 0.0238 |
| MAC→PRE | 0.004 | 0.0024 | 0.101 | 0.0597 |
| MAC→MAC_DEF [b] | 0.007 | 0.0014 | 0.173 | 0.0364 |
| MAC→MAC_MOR [b] | 0.004 | 0.0024 | 0.102 | 0.0596 |

[a] MEI→MEI_DEF and MEI→MEI_MOR was combined into MEI→DET in figure B1.

[b] MAC→MAC_DEF and MAC→MAC_MOR was combined into MAC→DET in figure B1.

**Table A8. Modeled carbon flows (mg C m$^{-2}$ d$^{-1}$) and comparison of POC partitioning.**

| Modeled flows | Definition | GC1 | GS1 | |
|---|---|---|---|---|
| POC$_{influx}$ | POC→DET | 2017.454 | 130.046 | |
| Export | DET→EXP | 1696.628 | 8.596 | |
| POC$_{retained}$ | POC$_{influx}$ − Export | 320.826 | 121.450 | This study |
| POC$_{burial}$ | DET→BUR | 248.475 | 108.872 | |
| POC$_{bio}$ | POC$_{influx}$ − Export − POC$_{burial}$ | 72.351 | 12.578 | |

| POC partitioning | Definition | GC1 | GS1 | Reference |
|---|---|---|---|---|
| Burial efficiency | POC$_{burial}$/POC$_{influx}$ | 12% | 84% | This study |
| | | 13% | | Hsu et al. (2014) |
| OC preservation efficiency | POC$_{burial}$/POC$_{retained}$ | 66[a]-77% | 69[a]-90% | This study |
| | | 70~100% | | Kao et al. (2014) |
| Export rate | Export/POC$_{influx}$ | 84% | 7% | This study |

| | | 82-87% | - | Huh et al. (2009) |
|---|---|---|---|---|
| POC into biological system | $POC_{bio}/POC_{influx}$ | 4% | 10% | This study |

[a] The difference between modeled TOU and measured TOU was added to $POC_{retained}$, and a more conservative OC
preservation efficiency was estimated.

**7 Appendice B**

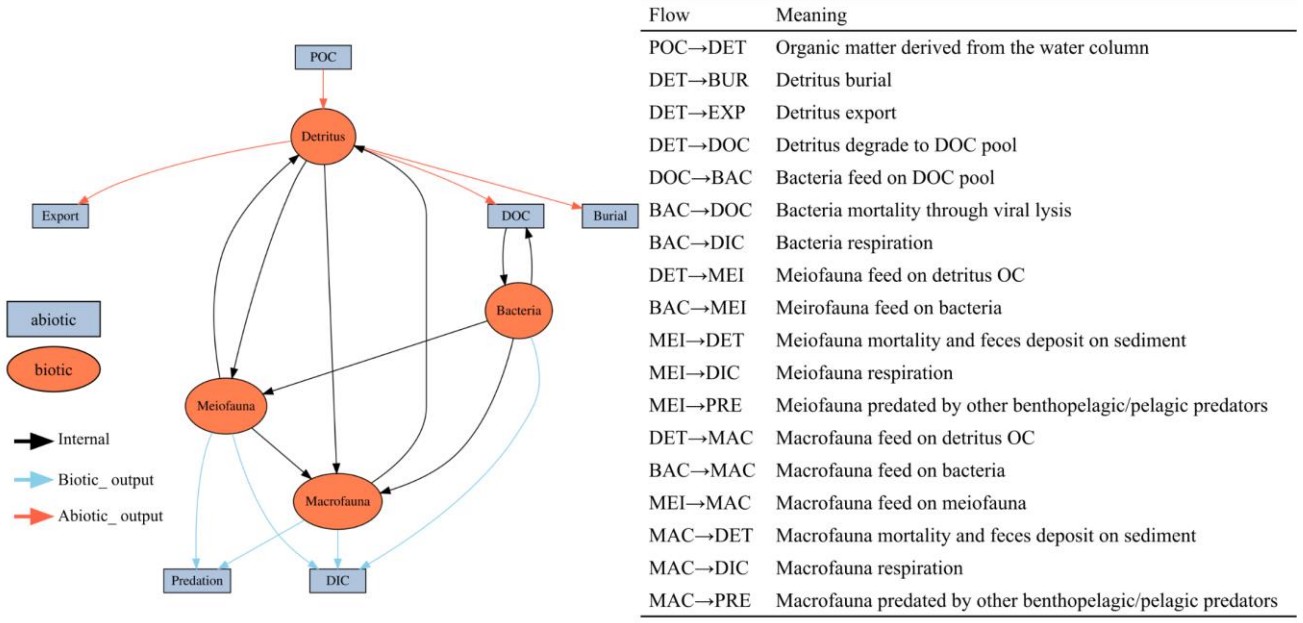

**Figure B1. The conceptual model of the food web structure formed the basis of our linear inverse model (LIM). The flows are with direction. For example, "SED→BAC" represents carbon flow from detritus to bacteria stock. See section "Food web structure" for further description.**

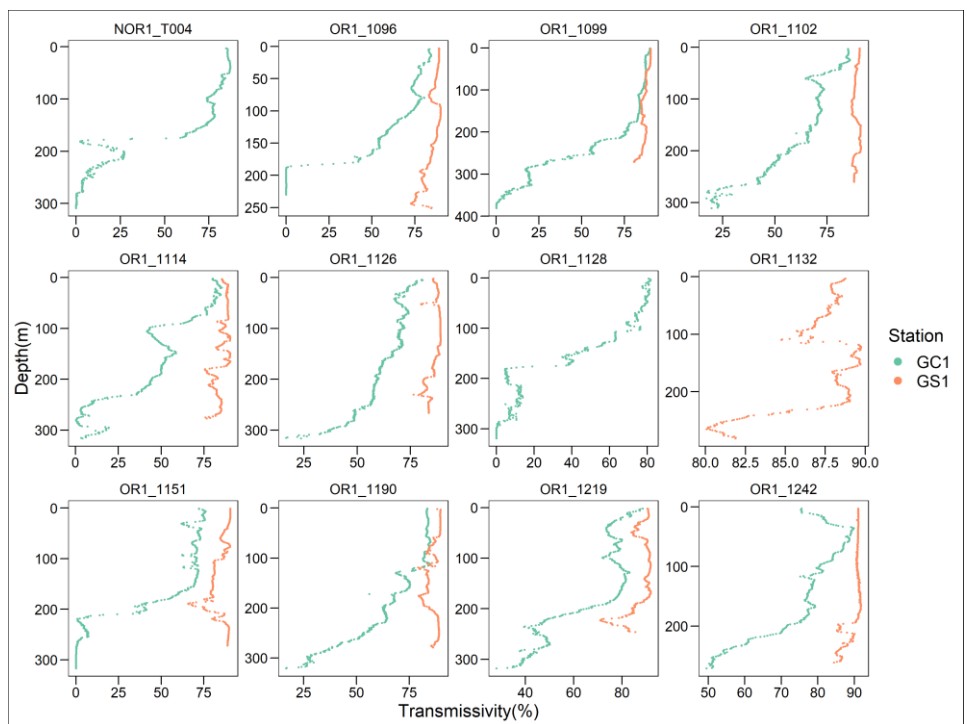

**Figure B2. Light transmission profile for each cruise of the two sites. Note that GC1 has a very low light transmission below 200 meter depth in almost all the sampling cruises.**

## 8 Code/Data availability

The datasets generated for this study can be found at the https://github.com/chueh1115/GPSC_LIM.

## 9 Author Contributions

J. -X. L. and C.-L. W. designed research; C.-C. T., Y.-S. L., J.-X. L., T.-H. T. and C.-L. W. performed research; C.-C. T. and C.-L. W. analyzed data; and C.-C. T., Y.-S. L., J.-X. L., T.-H. T., J. L., L.-H. L., P.-L. W., and C.-L. W. wrote the paper with intellectual contributions from all authors.

## 10 Competing interests

The authors declare that they have no conflict of interest.

## 11 Acknowledgments

We thank the Institute of Oceanography, National Taiwan University (NTU), and National Science and Technology Council (NSTC) for supporting the field works, analysis, and manuscript preparation. We also thank the captain, crew members, and technicians of the RV *Ocean Researcher I*, RV *New Ocean Researcher I*, and the graduate students who participated in the OCEAN 7090 Field Work in Marine Biology. This project is part of Fate of Terrestrial/Non-terrestrial

Sediments in High Yield Particle-Export River-sea Systems (FATES-HYPERS), sponsored by the NSTC 112-2611-M-002-011.

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
