# Peer review of "Contrasting carbon cycling in the benthic food webs between riverfed, high-energy canyon and upper continental slope"

_Biogeosciences, 2023_

## Author Comment (AC1)

Hendriks, A. J. (1999). Allometric scaling of rate, age and density parameters in ecological models. Oikos, 293-310.

Tenore, K. R. (1982). Comparison of the ecological energetics of the polychaetes Capitella capitata and Nereis succinea in experimental systems receiving similar levels of detritus. Netherlands Journal of Sea Research, 16, 46-54.

van Oevelen, D., Soetaert, K., & Heip, C. (2012). Carbon flows in the benthic food web of the Porcupine Abyssal Plain: The (un) importance of labile detritus in supporting microbial and faunal carbon demands. Limnology and Oceanography, 57(2), 645-664.

Middelboe, M., & Glud, R. N. (2006). Viral activity along a trophic gradient in continental margin sediments off central Chile. Marine Biology Research, 2(01), 41-51.

Danovaro, R., Dell'Anno, A., Corinaldesi, C., Magagnini, M., Noble, R., Tamburini, C., & Weinbauer, M. (2008). Major viral impact on the functioning of benthic deep-sea ecosystems. Nature, 454(7208), 1084-1087.

Van Oevelen, D., Soetaert, K., Garcia, R., De Stigter, H. C., Cunha, M. R., Pusceddu, A., & Danovaro, R. (2011). Canyon conditions impact carbon flows in food webs of three sections of the Nazaré canyon. Deep Sea Research Part II: Topical Studies in Oceanography, 58(23-24), 2461-2476.

Rowe, G.T., Wei, C.L., Nunnally, C., Haedrich, R., Montagna, P., Baguley, J.G., Bernhard, J.M., Wicksten, M., Ammons, A., Briones, E.E., Soliman, Y., Deming, J.W., 2008. Comparative biomass structure and estimated carbon flow in food webs in the deep Gulf of Mexico. Deep-Sea Research 55 (24–26), 2699–2711.

Middelburg, J. J., Vlug, T., Jaco, F., & Van der Nat, W. A. (1993). Organic matter mineralization in marine systems. Global and Planetary change, 8(1-2), 47-58.

Hedges, J. I., & Keil, R. G. (1995). Sedimentary organic matter preservation: an assessment and speculative synthesis. Marine chemistry, 49(2-3), 81-115.

Middelburg, J. J., & Meysman, F. J. (2007). Burial at sea. Science, 316(5829), 1294-1295.

---

## Author Response (AR1)

RC1: ['Comment on bg-2023-161'](), Anonymous Referee #1, 10 Nov 2023
The paper by Tung et al. presents a very valuable dataset of observations and modeling results of the benthic ecosystem of a submarine canyon and the continental slope of Taiwan/ South China Sea where such as study has not been performed before. This research therefore is of high interest to the broad marine community, after several major concerns have been addressed.

Author response: Thanks for the positive comments.

Major concerns:

1) The structure of the linear inverse model lacks mortality as part of physiological constraints. At the moment, no metazoan organism nor prokaryote deceases at the moment. The model compensates this by increased fluxes to the sediment which include both, feces and mortality, but it would be more accurate to include actual equations for mortality and properly constrain them.

Author response: The revised version of the model includes the metazoan mortality (see lines 118-119) with constraints from Hendriks (1999), Tenore (1982) and van Oevelen et al. (2012), and adequate equations from Middelboe and Glud (2006) and Danovaro et al. (2008) for constraining the microbial loop (see Table 1).

2) The authors lacked parts of the microbial loop. Like mentioned above, the bacteria do not 'die' which would be caused by virus-induced lysis and it is highly unlikely/ impossible, that they directly feed on sediment. Instead an degradation step of detritus to DOC which then is taken up by bacteria should be included. Dying bacteria also contribute to the DOC pool. Implementing the microbial loop (check papers about LIM in the deep sea by e.g. van Oevelen, Dunlop, Durden, Stratmann, etc.) will not only improve the model quality/ present the natural environment more appropriately, it also allows to compare modeling results of this study with other LIM studies.

Author response: The revised version of the model includes the microbial loop, including the process of detritus degradation to DOC, DOC uptake by prokaryotes, viral-induced prokaryotes mortality and its contribution to DOU, and metazoan feeding on prokaryotes (see Figure 4, lines 114-116 in methods, lines 341-342 in results, and lines 549—553 in discussion). Furthermore, we added Figure 5 to illustrate the fate of secondary production of each biotic compartment to make a comparison with van Oevelen et al. (2011) in section 4.3.

3) Why did the authors pool all metazoans in two size classes without splitting them into larger groups or feeding types? This should either be done or be discussed as a major limitation of study in the corresponding section of the discussion. At the moment, it is very much focused on the sampling and the report of the limitations of the study is underdeveloped.

Author response: We added a paragraph in section 4.1 (see lines 423-435), reading as "In food web studies, the general trophic structure of communities is established by grouping organisms into broader feeding types based on information gathered from the literature (e.g., Fauchald and Jumars 1979; Lincoln 1979; Dauwe et al. 1998). However, the lack of adequate taxonomic resolution (e.g., family, genus, or species level) hinders our ability to segregate data into functional groups. Consequently, we derived estimates of the metazoan carbon flows solely based on physiological constraints and mass balancing rather than relying on the feeding preferences of the functional groups. Moreover, some metazoans can switch their feeding behavior in response to changing food quality and quantity in the environment. For instance, some Polychaeta and Mollusca species can transition from deposit-feeding to suspension-feeding or vice versa following fluctuations in the flux of suspended particles (Taghon and Greene, 1992), providing an advantage in environments with varying flow velocities. A mesocosm observational study also demonstrated that two surface deposit feeder species, *Sipunculus* sp. (Sipuncula) and *Spiophanes kroeyeri* (Polychaeta), shifting from deposit-feeding to suspension-feeding under different flow conditions (Thomsen and Flach, 1997). Thus, while aggregating all metazoans into two size classes oversimplifies the food web model, it may help prevent misrepresenting of the relative contributions of taxa exhibiting high feeding plasticity."

4) Going in the same direction: Why did the authors use only one detritus pool instead of splitting it up in several pools like presented in studies by van Oevelen, Dunlop, Durden, or Stratmann? This should be discussed in the model limitation section.

Author response: We added a paragraph in section 4.1 (see lines 436-447), reading as "On the other hand, the organic matter degradation in sediments exhibits high variability, influenced by factors such as organic matter chemistry, sediment physical characteristics, and biological agents involved in decomposition (Middelburg and Meysman, 2007). As a result, the proportions of detrital organic carbon components (i.e., labile, semi-labile, and refractory fractions) typically differ across study locations and cannot be directly inferred from previous literature (e.g., van Oevelen et al. 2011; Dunlop et al. 2016; Stratmann et al. 2018; Durden et al. 2020). Due to lacking

empirical data, we aggregated all detritus into a single compartment, consistent with the approach in other benthic food web model studies (e.g., Rowe et al. 2008). Additionally, organic matter degradation occurs over a broad spectrum of time scales, from minutes for biochemical breakdown in animal guts to $10^6$ years for organic carbon mineralization in deep-sea sediments (Middelburg et al., 1993). This wide dynamic range underscores the significant influence of the characteristic time scale of experimental observations on measurable degradation rates (Hedges and Keil, 1995). Given that our model measures sediment organic carbon oxidation through oxygen consumption, it likely emphasizes the estimation of labile organic carbon degradation over semi-labile and refractory fractions due to their extended time scales."

5) The authors should also discuss how the exclusion of megabenthos affects the results of the study, especially because van Oevelen et al. (2011) showed the importance of megabenthos at parts of the Nazare canyon.

Author response: We revised the paragraph in section 4.1 (see lines 448-461), reading as "The current food web models omit megafauna stock due to data limitation, notably the technical challenges of trawling in hazardous areas like the GPSC. Consequently, the megafauna predation is only constrained by the net growth efficiency of meiofauna and macrofauna, with all metazoan growths assumed to be consumed by megafauna. In the deep sea, the megafauna abundance and biomass are generally lower and decrease more rapidly with depth than the smaller infauna (Rex et al., 2006; Wei et al., 2010). However, certain areas can exhibit high megafauna and fish densities (Sibuet, 1977; Hecker, 1994; Fodrie et al., 2009), which can influence the redistribution and quality of OM in the marine sediments (Smallwood et al., 1999), potentially affecting the food web dynamics. Another consequence of the current food web is the lack of detrital carbon consumption by deposit-feeding megafauna. While van Oevelen et al. (2011) highlighted the significance of deposit-feeding megabenthos at the mid-section of the Nazaré Canyon, their sampling depths are considerably deeper than ours. It is suggested that the intense flow regimes at the canyon head (i.e., tidal and turbidity currents) may favor the mobile megafauna (i.e., fish, crabs, and octopus) over the deposit-feeding counterparts (i.e., sea urchins) because the mobile megafauna predators are better coping with physical disturbance and may benefit from transient food subsidy in the canyon (Vetter and Dayton, 1999; Vetter et al., 2010). Per the presented model structure, the sediment detrital carbon in GPSC would be directed toward export, burial, microbial loop, and smaller metazoan. Nonetheless, the absence of depositfeeding megafauna is not expected to disproportionately influence the GPSC food web."

-> Summaryzing these comments, I believe that the authors should re-develop the linear inverse models for the two stations to implement my concerns 1 and 2 in the model. The other aspects should preferrably also be included in the new versions of the model, but if that is not possible, they have to be elaborated in the discussion section.

Author response: We revised the model by considering the metazoan mortality and adding the microbial loop. We discussed the limitations, including splitting all metazoans into two groups, using only one detritus pool, and the effect of exclusion of megabenthos.

Minor/ Technical comments:

l 80: Avoid abbreviations in headlines if not strictly necessary. Hence, please write the full name of the canyons here.

Author response: Revised.

l 111: Here and in other parts of the manuscript, the authors refer to Fig. B1. I think they refer to Fig. 4, but I'm not sure. Either way, they should correct this and refer to the correct figure.

Author response: Here, we indeed refer to Fig. B1. We intend to use Fig. B1 to explain the food web structure with a table of figure legends describing the definition of each flow.

l 112: "export processes" What export processes do the authors have in mind here? Please explain in a short half sentence.

Author response: Done. The sentence now reads (see lines 111-113), "In the model, the influx of POC consists of a mixture of OC derived from the water column, with a portion exiting the network through burial and export processes such as removal from sediment surface or resuspension by bottom currents (represented by orange flows)."

l 145: Which "taxonomic group" did the authors include here? Please be more specific (which taxonomic rank, or list the taxonomic groups).

Author response: Done. We add a sentence (lines 150-151), "Both the macrofauna and meiofauna were identified to the lowest possible taxonomic level (phylum, class or order)."

Figures: I don't know how it will look like in the published manuscript, but at the moment the letter size of the figures is relative/ very small. The authors should increase the letter size in the figures a bit.

Author response: We have increased the letter size in the figure.

Citation: https://doi.org/10.5194/bg-2023-161-RC1

RC2: 'Comment on bg-2023-161', Anonymous Referee #2, 01 Dec 2023
Tung et al., "Contrasting carbon cycling in the benthic food webs between river-fed, high-energy canyon and upper continental slope."

Tung et al. present a nice study investigating organic carbon cycling across the sediment-water interface in Gaoping Submarine Canyon off Southwest Taiwan. This study is notable in how they combined sampling of the benthic food web over multiple seasons with a linear inverse model to investigate the fate of organic carbon and the role of benthic communities in modulating carbon flows in two distinct sites, representing a high energy, high disturbance site with low faunal biomass vs a more stable site with higher faunal biomass. The paper is overall well written, with a nice discussion, particularly on limitations of the LIM model in estimating the food web consumption of oxygen.

Author response: Thanks for the encouraging words.

I have two main comments:

1) Since the wet and dry seasons are so different, and the study only sampled once during the summer, what would the results of the LIM be if you separated out the seasons? E.g., one model result for spring, and one for autumn? Or just dry season only (grouping together spring and autumn and excluding summer)? I'm not sure it makes sense to include the summer data for an "annual average," as the data become quite skewed towards the dry season. Further, the TOU and DOU measurements are only available for the spring and autumn, and thus wouldn't be representative of an annual average anyway.

Author response: as the reviewer suggests, since we only sampled once in summer, and the OU measurements were only available for the dry periods (i.e., spring and autumn), the revised version of the model used dry period-only data (see lines 124-125 in methods; lines 285-287, lines 305-307, and lines 312-313 in results; lines 411-413 in discussion).

2) I personally found the network characteristics hard to understand, as someone who works on food web dynamics but not on deep sea food webs. There was no definitions or equations describing the main terms used, and so the whole set of analyses on the food web network was lost to me. If possible, please include in the supplemental a set of definitions and/or equations describing the terms that are used. It would just help broaden the readership of this manuscript.

Author response: Thanks for the suggestion. We include a more detailed description and equations of each network index in the supplemental information (see lines 255-258, Table A3, and A4).

Minor comments:

Fig 4. I found the overlapping numbers and arrows, particularly in the POC -> Sediment arrow hard to read. Also, would it be possible to include in the table or the supplement the fraction (or percentage) of flows going to each sink? That would help readers be able to more easily interpret how the ecosystem functions differently in the two sites.

Author response: We revised the overlapping numbers in Fig. 4 by changing the color and text positions. Instead of the fractions of carbon flows going to each sink, we added Figure 5 to illustrate the fate of secondary production of each biotic compartment (i.e., the fractions of carbon flows leaving prokaryote, meiofauna, and megafauna). We think this is better illustrated and can make a comparison with van Oevelen et al. (2011) in section 4.3.

Section 3.7, line 368. Total system throughflow is marked as T.. and total system throughput is marked as TST. I believe this should be flipped.

Author response: Revised.

**Citation**: https://doi.org/10.5194/bg-2023-161-RC2